# Self-Supervised Learning Disentangled Group Representation as Feature

**Tan Wang**[1] **Zhongqi Yue**[1,3] **Jianqiang Huang**[1,3] **Qianru Sun**[2] **Hanwang Zhang**[1]

[1]Nanyang Technological University  [2]Singapore Management University  [3]Damo Academy, Alibaba Group

{tan317,yuez0003,hanwangzhang}@ntu.edu.sg

jianqiang.jqh@gmail.com  qianrusun@smu.edu.sg

## Abstract

A good visual representation is an inference map from observations (images) to features (vectors) that faithfully reflects the hidden modularized generative factors (semantics). In this paper, we formulate the notion of "good" representation from a group-theoretic view using Higgins' definition of ***disentangled representation*** [40], and show that existing Self-Supervised Learning (SSL) only disentangles simple augmentation features such as rotation and colorization, thus unable to modularize the remaining semantics. To break the limitation, we propose an iterative SSL algorithm: Iterative Partition-based Invariant Risk Minimization (IP-IRM), which successfully grounds the abstract semantics and the group acting on them into concrete contrastive learning. At each iteration, IP-IRM first partitions the training samples into two subsets that correspond to an entangled group element. Then, it minimizes a subset-invariant contrastive loss, where the invariance guarantees to disentangle the group element. We prove that IP-IRM converges to a fully disentangled representation and show its effectiveness on various benchmarks. Codes are available at https://github.com/Wangt-CN/IP-IRM.

## 1 Introduction

Deep learning is all about learning feature representations [5]. Compared to the conventional end-to-end supervised learning, Self-Supervised Learning (SSL) first learns a generic feature representation (*e.g.*, a network backbone) by training with unsupervised pretext tasks such as the prevailing contrastive objective [36, 16], and then the above stage-1 feature is expected to serve various stage-2 applications with proper fine-tuning. SSL for visual representation is so fascinating that it is the first time that we can obtain "good" visual features for free, just like the trending pre-training in NLP community [26, 8]. However, most SSL works only care how much stage-2 performance an SSL feature can improve, but overlook what feature SSL is learning, why it can be learned, what cannot be learned, what the gap between SSL and Supervised Learning (SL) is, and when SSL can surpass SL?

The crux of answering those questions is to formally understand *what a feature representation is* and *what a good one is*. We postulate the classic world model of visual generation and feature representation [1, 69]

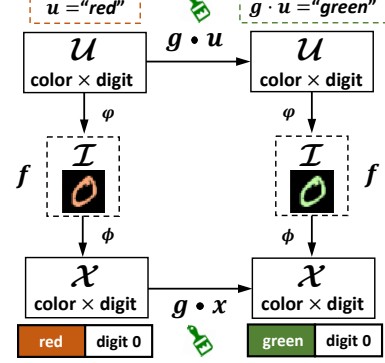

Figure 1: Disentangled representation is an equivariant map between the semantic space $\mathcal{U}$ and the vector space $\mathcal{X}$, which is decomposed into "color" and "digit".

as in Figure 1. Let $\mathcal{U}$ be a set of (unseen) *semantics*, *e.g.*, attributes such as "digit" and "color". There

35th Conference on Neural Information Processing Systems (NeurIPS 2021).

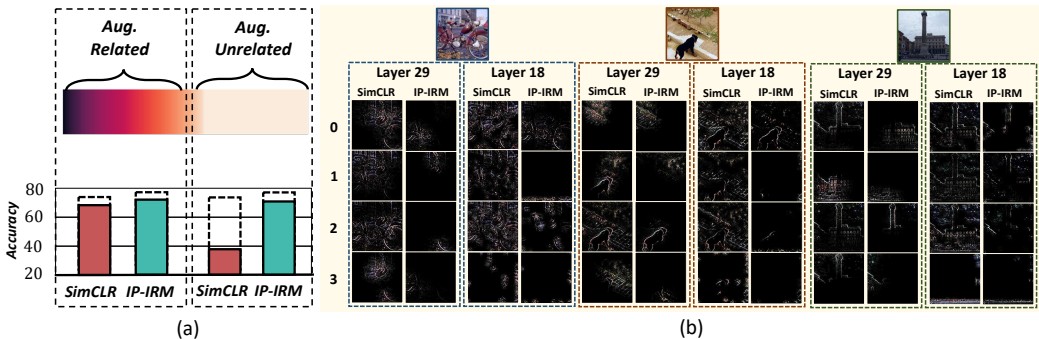

Figure 2: (a) The heat map visualizes feature dimensions related to augmentations (aug. related) and unrelated to augmentations (aug. unrelated), whose respective classification accuracy is shown in the bar chart below. Dashed bar denotes the accuracy using full feature dimensions. Experiment was performed on STL10 [22] with representation learnt with SimCLR [16] and our IP-IRM. (b) Visualization of CNN activations [77] of 4 filters on layer 29 and 18 of VGG [75] trained on ImageNet100 [81]. The filters were chosen by first clustering the aug. unrelated filters with $k$-means ($k = 4$) and then selecting the filters corresponding to the cluster centers.

is a set of *independent and causal mechanisms* [66] $\varphi : \mathcal{U} \rightarrow \mathcal{I}$, generating images from semantics, *e.g.*, writing a digit "0" when thinking of "0" [74]. A **visual representation** is the inference process $\phi : \mathcal{I} \rightarrow \mathcal{X}$ that maps image pixels to vector space features, *e.g.*, a neural network. We define **semantic representation** as the functional composition $f : \mathcal{U} \rightarrow \mathcal{I} \rightarrow \mathcal{X}$. In this paper, we are only interested in the parameterization of the inference process for feature extraction, but not the generation process, *i.e.*, we assume $\forall I \in \mathcal{I}, \exists u \in \mathcal{U}$, such that $I = \varphi(u)$ is fixed as the observation of each image sample. Therefore, we consider semantic and visual representations the same as **feature representation**, or simply **representation**, and we slightly abuse $\phi(I) := f\left(\varphi^{-1}(I)\right)$, *i.e.*, $\phi$ and $f$ share the same trainable parameters. We call the vector $\mathbf{x} = \phi(I)$ as **feature**, where $\mathbf{x} \in \mathcal{X}$.

We propose to use Higgins' definition of **disentangled representation** [40] to define what is "good".

**Definition 1.** (Disentangled Representation) *Let $\mathcal{G}$ be the group acting on $\mathcal{U}$, i.e., $g \cdot u \in \mathcal{U} \times \mathcal{U}$ transforms $u \in \mathcal{U}$, e.g., a "turn green" group element changing the semantic from "red" to "green". Suppose there is a direct product decomposition[1] $\mathcal{G} = g_1 \times \ldots \times g_m$ and $\mathcal{U} = \mathcal{U}_1 \times \ldots \times \mathcal{U}_m$, where $g_i$ acts on $\mathcal{U}_i$ respectively. A feature representation is disentangled if there exists a group $\mathcal{G}$ acting on $\mathcal{X}$ such that:*

1. *Equivariant: $\forall g \in \mathcal{G}, \forall u \in \mathcal{U}, f(g \cdot u) = g \cdot f(u)$, e.g., the feature of the changed semantic: "red" to "green" in $\mathcal{U}$, is equivalent to directly change the color vector in $\mathcal{X}$ from "red" to "green".*

2. *Decomposable: there is a decomposition $\mathcal{X} = \mathcal{X}_1 \times \ldots \times \mathcal{X}_m$, such that each $\mathcal{X}_i$ is fixed by the action of all $g_j, j \neq i$ and affected only by $g_i$, e.g., changing the "color" semantic in $\mathcal{U}$ does not affect the "digit" vector in $\mathcal{X}$.*

Compared to the previous definition of feature representation which is a static mapping, the disentangled representation in Definition 1 is dynamic as it explicitly incorporate **group representation** [35], which is a homomorphism from group to group actions on a space, *e.g.*, $\mathcal{G} \rightarrow \mathcal{X} \times \mathcal{X}$, and it is common to use the feature space $\mathcal{X}$ as a shorthand—this is where our title stands.

Definition 1 defines "good" features in the common views: 1) *Robustness*: a good feature should be invariant to the change of environmental semantics, such as external interventions [45, 87] or domain shifts [32]. By the above definition, a change is always retained in a subspace $\mathcal{X}_i$, while others are not affected. Hence, the subsequent classifier will focus on the invariant features and ignore the ever-changing $\mathcal{X}_i$. 2) *Zero-shot Generalization*: even if a new combination of semantics is unseen in training, each semantic has been learned as features. So, the metrics of each $\mathcal{X}_i$ trained by seen samples remain valid for unseen samples [95].

Are the existing SSL methods learning disentangled representations? No. We show in Section 4 that they can only disentangle representations according to the hand-crafted augmentations, *e.g.*, color jitter and rotation. For example, in Figure 2 (a), even if we only use the augmentation-related feature,

---

[1]Note that $g_i$ can also denote a cyclic subgroup $\mathcal{G}_i$ such as rotation $[0° : 1° : 360°]$, or a countable one but treated as cyclic such as translation $[(0, 0) : (1, 1) : (\text{width}, \text{height})]$ and color $[0 : 1 : 255]$.

the classification accuracy of a standard SSL (SimCLR [16]) does not lose much as compared to the full feature use. Figure 2 (b) visualizes that the CNN features in each layer are indeed entangled (*e.g.*, tyre, motor, and background in the motorcycle image). In contrast, our approach IP-IRM, to be introduced below, disentangles more useful features beyond augmentations.

In this paper, we propose Iterative Partition-based Invariant Risk Minimization (**IP-IRM** [ˌaiˈpəːm]) that guarantees to learn disentangled representations in an SSL fashion. We present the algorithm in Section 3, followed by the theoretical justifications in Section 4. In a nutshell, at each iteration, IP-IRM first partitions the training data into two disjoint subsets, each of which is an orbit of the already disentangled group, and the cross-orbit group corresponds to an entangled group element $g_i$. Then, we adopt the **Invariant Risk Minimization (IRM)** [2] to implement a **partition-based** SSL, which disentangles the representation $\mathcal{X}_i$ *w.r.t.* $g_i$. Iterating the above two steps eventually converges to a fully disentangled representation *w.r.t.* $\prod_{i=1}^{m} g_i$. In Section 5, we show promising experimental results on various feature disentanglement and SSL benchmarks.

## 2 Related Work

**Self-Supervised Learning**. SSL aims to learn representations from unlabeled data with hand-crafted pretext tasks [28, 63, 33]. Recently, Contrastive learning [65, 61, 38, 80, 16] prevails in most state-of-the-art methods. The key is to map positive samples closer, while pushing apart negative ones in the feature space. Specifically, the positive samples are from the augmented views [82, 3, 94, 42] of each instance and the negative ones are other instances. Along this direction, follow-up methods are mainly four-fold: 1) Memory-bank [90, 61, 36, 18]: storing the prototypes of all the instances computed previously into a memory bank to benefit from a large number of negative samples. 2) Using siamese network [7] to avoid representation collapse [34, 19, 83]. 3) Assigning clusters to samples to integrate inter-instance similarity into contrastive learning [11, 12, 13, 88, 56]. 4) Seeking hard negative samples with adversarial training or better sampling strategies [73, 20, 44, 48]. In contrast, our proposed IP-IRM jumps out of the above frame and introduces the *disentangled representation* into SSL with group theory to show the limitations of existing SSL and how to break through them.

**Disentangled Representation**. This notion dates back to [4], and henceforward becomes a high-level goal of separating the factors of variations in the data [84, 79, 86, 58]. Several works aim to provide a more precise description [27, 29, 72] by adopting an information-theoretic view [17, 27] and measuring the properties of a disentangled representation explicitly [29, 72]. We adopt the recent group-theoretic definition from Higgins *et al.* [40], which not only unifies the existing, but also resolves the previous controversial points [78, 59]. Although supervised learning of disentangled representation is a well-studied field [100, 43, 10, 70, 49], unsupervised disentanglement based on GAN [17, 64, 57, 71] or VAE [39, 15, 99, 50] is still believed to be theoretically challenging [59]. Thanks to the Higgins' definition, we prove that the proposed IP-IRM converges with full-semantic disentanglement using group representation theory. Notably, IP-IRM learns a disentangled representation with an inference process, without using generative models as in all the existing unsupervised methods, making IP-IRM applicable even on large-scale datasets.

**Group Representation Learning**. A group representation has two elements [47, 35]: 1) a homomorphism (*e.g.*, a mapping function) from the group to its group action acting on a vector space, and 2) the vector space. Usually, when there is no ambiguity, we can use either element as the definition. Most existing works focus on learning the first element. They first define the group of interest, such as spherical rotations [24] or image scaling [89, 76], and then learn the parameters of the group actions [23, 46, 68]. In contrast, we focus on the second element; more specifically, we are interested in learning a map between two vector spaces: image pixel space and feature vector space. Our representation learning is flexible because it delays the group action learning to downstream tasks on demand. For example, in a classification task, a classifier can be seen as a group action that is invariant to class-agnostic groups but equivariant to class-specific groups (see Section 4).

## 3 IP-IRM Algorithm

**Notations**. Our goal is to learn the feature extractor $\phi$ in a self-supervised fashion. We define a partition matrix $\mathbf{P} \in \{0, 1\}^{N \times 2}$ that partitions $N$ training images into 2 disjoint subsets. $P_{i,k} = 1$ if the $i$-th image belongs to the $k$-th subset and 0 otherwise. Suppose we have a pretext task loss

function $\mathcal{L}(\phi, \theta = 1, k, \mathbf{P})$ defined on the samples in the $k$-th subset, where $\theta = 1$ is a "dummy" parameter used to evaluate the invariance of the SSL loss across the subsets (later discussed in Step 1). For example, $\mathcal{L}$ can be defined as:

$$\mathcal{L}(\phi, \theta = 1, k, \mathbf{P}) = \sum_{\mathbf{x} \in \mathcal{X}_k} -\log \frac{\exp\left(\mathbf{x}^T \mathbf{x}^* \cdot \theta\right)}{\sum_{\mathbf{x}' \in \mathcal{X}_k \cup \mathcal{X}^* \setminus \mathbf{x}} \exp\left(\mathbf{x}^T \mathbf{x}' \cdot \theta\right)}, \quad (1)$$

where $\mathcal{X}_k = \phi(\{I_i | P_{i,k} = 1\})$, and $\mathbf{x}^* \in \mathcal{X}^*$ is the augmented view feature of $\mathbf{x} \in \mathcal{X}_k$.

**Input**. $N$ training images. Randomly initialized $\phi$. A partition matrix $\mathbf{P}$ initialized such that the first column of $\mathbf{P}$ is 1, *i.e.*, all samples belong to the first subset. Set $\mathcal{P} = \{\mathbf{P}\}$.

**Output**. Disentangled feature extractor $\phi$.

**Step 1 [Update $\phi$]**. We update $\phi$ by:

$$\min_{\phi} \sum_{\mathbf{P} \in \mathcal{P}} \sum_{k=1}^{2} \left[ \mathcal{L}(\phi, \theta = 1, k, \mathbf{P}) + \lambda_1 \|\nabla_{\theta=1} \mathcal{L}(\phi, \theta = 1, k, \mathbf{P})\|^2 \right], \quad (2)$$

where $\lambda_1$ is a hyper-parameter. The second term delineates how far the contrast in one subset is from a constant baseline $\theta = 1$. The minimization of both of them encourages $\phi$ in different subsets close to the same baseline, *i.e.*, invariance across the subsets. See IRM [2] for more details. In particular, the first iteration corresponds to the standard SSL with $\mathcal{X}_1$ in Eq. (1) containing all training images.

**Step 2 [Update P]**. We fix $\phi$ and find a new partition $\mathbf{P}^*$ by

$$\mathbf{P}^* = \arg\max_{\mathbf{P}} \sum_{k=1}^{2} \left[ \mathcal{L}(\phi, \theta = 1, k, \mathbf{P}) + \lambda_2 \|\nabla_{\theta=1} \mathcal{L}(\phi, \theta = 1, k, \mathbf{P})\|^2 \right], \quad (3)$$

where $\lambda_2$ is a hyper-parameter. In practice, we use a continuous partition matrix in $\mathbb{R}^{N \times 2}$ during optimization and then threshold it to $\{0, 1\}^{N \times 2}$.

We update $\mathcal{P} \leftarrow \mathcal{P} \cup \mathbf{P}^*$ and iterate the above two steps until convergence.

## 4 Justification

Recall that IP-IRM uses training sample **partitions** to learn the disentangled representations *w.r.t.* $\prod_{i=1}^{m} g_i$. As we have a $\mathcal{G}$-equivariant feature map between the sample space $\mathcal{I}$ and feature space $\mathcal{X}$ (the equivariance is later guaranteed by Lemma 1), we slightly abuse the notation by using $\mathcal{X}$ to denote both spaces. Also, we assume that $\mathcal{X}$ is a **homogeneous** space of $\mathcal{G}$, *i.e.*, any sample $\mathbf{x}' \in \mathcal{X}$ can be transited from another sample $\mathbf{x}$ by a group action $g \cdot \mathbf{x}$. Intuitively, $\mathcal{G}$ is all you need to describe the diversity of the training set. It is worth noting that $g$ is any group element in $\mathcal{G}$ while $g_i$ is a Cartesian "building block" of $\mathcal{G}$, *e.g.*, $g$ can be decomposed by $(g_1, g_2, ..., g_m)$.

We show that partition and group are tightly connected by the concept of **orbit**. Given a sample $\mathbf{x} \in \mathcal{X}$, its group orbit *w.r.t.* $\mathcal{G}$ is a sample set $\mathcal{G}(\mathbf{x}) = \{g \cdot \mathbf{x} \mid g \in \mathcal{G}\}$. As shown in Figure 3 (a), if $\mathcal{G}$ is a set of attributes shared by classes, *e.g.*, "color" and "pose", the orbit is the sample set of the class of $\mathbf{x}$; in Figure 3 (b), if $\mathcal{G}$ denotes augmentations, the orbit is the set of augmented images. In particular, we can see that the disjoint orbits in Figure 3 naturally form a partition. Formally, we have the following definition:

**Definition 2.** (Orbit & Partition [47]) *Given a subgroup $\mathcal{D} \subset \mathcal{G}$, it partitions $\mathcal{X}$ into the disjoint subsets: $\{\mathcal{D}(c_1 \cdot \mathbf{x}), ..., \mathcal{D}(c_k \cdot \mathbf{x})\}$, where $k$ is the number of cosets $\{c_1 \mathcal{D}, ..., c_k \mathcal{D}\}$, and the cosets form a factor group[1] $\mathcal{G}/\mathcal{D} = \{c_i\}_{i=1}^{k}$. In particular, $c_i \cdot \mathbf{x}$ can be considered as a sample of the $i$-th class, transited from any sample $\mathbf{x} \in \mathcal{X}$.*

Interestingly, the partition offers a new perspective for the training data format in Supervised Learning (SL) and Self-Supervised Learning (SSL). In SL, as shown in Figure 3 (a), the data is labeled with $k$ classes, each of which is an orbit with $\mathcal{D}(c_i \cdot \mathbf{x})$ training samples, whose variations are depicted

---

[1]Given $\mathcal{G} = \mathcal{D} \times \mathcal{K}$ with $\mathcal{K} = c_1 \times \ldots \times c_k$, then $\bar{\mathcal{D}} = \{(d, e) \mid d \in \mathcal{D}\}$ is a normal subgroup of $\mathcal{G}$, and $\mathcal{G}/\bar{\mathcal{D}}$ is isomorphic to $\mathcal{K}$ [47]. We write $\mathcal{G}/\mathcal{D} = \{c_i\}_{i=1}^{k}$ with slight abuse of notation.

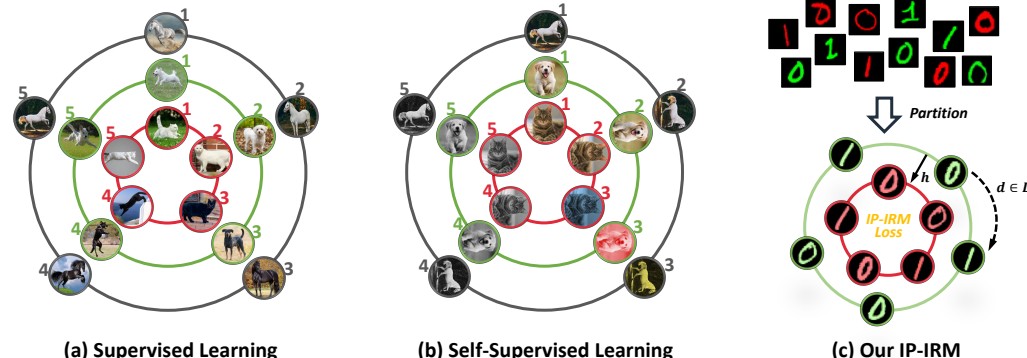

**(a) Supervised Learning**  **(b) Self-Supervised Learning**  **(c) Our IP-IRM**

Figure 3: Each orbit only illustrates with 5 samples. (a) Orbit: the training samples of a class; $\mathcal{D}$ in-orbit actions: intra-class variations (1→2:standing, 2→3:blacken, 3→4:jumping, 4→5:whiten, 5→1:running); $\mathcal{G}/\mathcal{D}$ cross-orbit actions: inter-class variations. (b) Orbit: a sample and its augmented samples; $\mathcal{D}$ in-orbit actions: augmentations (1→2:clock-wise rotation, 2→3:color jitter, 3→4:gray scale, 4→5:counterclockwise rotation, 5→1:color); $\mathcal{G}/\mathcal{D}$ cross-orbit actions: inter-sample variations. (c) Step 2 in IP-IRM discovers 2 orbits, where the cross-orbit action corresponds to a group action "green to red" or "red to green", which is yet disentangled.

by the class-sharing attribute group $\mathcal{D}$. The cross-orbit group action, *e.g.*, $c_{\text{dog}} \cdot \mathbf{x}$, can be read as "turn $\mathbf{x}$ into a dog" and such "turn" is always valid due to the assumption that $\mathcal{X}$ is a homogeneous space of $\mathcal{G}$. In SSL, as shown in Figure 3 (b), each training sample $\mathbf{x}$ is augmented by the group $\mathcal{D}$. So, $\mathcal{D}(c_i \cdot \mathbf{x})$ consists of all the augmentations of the $i$-th sample, where the cross-orbit group action $c_i \cdot \mathbf{x}$ can be read as "turn $\mathbf{x}$ into the $i$-th sample".

Thanks to the orbit and partition view of training data, we are ready to revisit model **generalization** in a group-theoretic view by using **invariance** and **equivariance**—the two sides of the coin, whose name is **disentanglement**. For SL, we expect that a good feature is disentangled into a class-agnostic part and a class-specific part: the former (latter) is invariant (equivariant) to $\mathcal{G}/\mathcal{D}$—cross-orbit traverse, but equivariant (invariant) to $\mathcal{D}$—in-orbit traverse. By using such feature, a model can generalize to diverse testing samples (limited to $|\mathcal{D}|$ variations) by only keeping the class-specific feature. Formally, we prove that we can achieve such disentanglement by contrastive learning:

**Lemma 1.** (Disentanglement by Contrastive Learning) *Training loss* $-\log \frac{\exp(\mathbf{x}_i^T \mathbf{x}_j)}{\sum_{\mathbf{x} \in \mathcal{X}} \exp(\mathbf{x}_j^T \mathbf{x})}$ *disentangles* $\mathcal{X}$ *w.r.t.* $(\mathcal{G}/\mathcal{D}) \times \mathcal{D}$, *where* $\mathbf{x}_i$ *and* $\mathbf{x}_j$ *are from the same orbit.*

We can draw the following interesting corollaries from Lemma 1 (details in Appendix):

1. If we use all the samples in the denominator of the loss, we can approximate to $\mathcal{G}$-equivariant features given limited training samples. This is because the loss minimization guarantees $\forall (\mathbf{x}_i, \mathbf{x}_j) \in \mathcal{X} \times \mathcal{X}, i \neq j \rightarrow \mathbf{x}_i \neq \mathbf{x}_j$, *i.e.*, any pair corresponds to a group action.

2. Conventional cross-entropy loss in SL is a special case, if we define $\mathbf{x} \in \mathcal{X} = \{\mathbf{x}_1, ..., \mathbf{x}_k\}$ as $k$ classifier weights. So, SL does not guarantee the disentanglement of $\mathcal{G}/\mathcal{D}$, which causes generalization error if the class domain of downstream task is different from SL pre-training, *e.g.*, a subset of $\mathcal{G}/\mathcal{D}$.

3. In contrastive learning based SSL, $\mathcal{D}$ = "augmentations" (recall Figure 2), and the number of augmentations $|\mathcal{D}_{\text{aug}}|$ is generally much smaller compared to the class-wise sample diversity $|\mathcal{D}_{\text{SL}}|$ in SL. This enables the SL model to generalize to more diverse testing samples ($|\mathcal{D}_{\text{SL}}|$) by filtering out the class-agnostic features (*e.g.*, background) and focusing on the class-specific ones (*e.g.*, foreground), which explains why SSL is worse than SL in downstream classification.

4. In SL, if the number of training samples per orbit is not enough, *i.e.*, smaller than $|\mathcal{D}(c_i \cdot \mathbf{x})|$, the disentanglement between $\mathcal{D}$ and $\mathcal{G}/\mathcal{D}$ cannot be guaranteed, such as the challenges in few-shot learning [96]. Fortunately, in SSL, the number is enough as we always include all the augmented samples in training. Moreover, we conjecture that $\mathcal{D}_{\text{aug}}$ only contains simple cyclic group elements such as rotation and colorization, which are easier for representation learning.

Lemma 1 does not guarantee the decomposability of each $d \in \mathcal{D}$. Nonetheless, the downstream model can still generalize by keeping the class-specific features affected by $\mathcal{G}/\mathcal{D}$. Therefore, the key to fill the gap or even let SSL surpass SL is to achieve the full disentanglement of $\mathcal{G}/\mathcal{D}_{\text{aug}}$.

| | Method | DCI | IRS | MOD | EXP | LR | GBT | Average |
|---|---|---|---|---|---|---|---|---|
| CMNIST | VAE [51] | **0.948**±0.004 | - | 0.664±0.121 | 0.968±0.007 | 0.824±0.019 | **0.948**±0.004 | 0.849±0.057 |
| | $\beta$-VAE [41] | 0.945±0.002 | - | 0.705±0.073 | 0.963±0.006 | 0.809±0.013 | 0.945±0.003 | 0.874±0.015 |
| | $\beta$-AnnealVAE [9] | 0.911±0.002 | - | 0.790±0.075 | 0.965±0.007 | 0.821±0.022 | 0.911±0.002 | 0.880±0.016 |
| | $\beta$-TCVAE [15] | 0.914±0.008 | - | 0.864±0.095 | 0.962±0.010 | 0.801±0.024 | 0.914±0.008 | 0.891±0.014 |
| | Factor-VAE [50] | 0.916±0.004 | - | **0.893**±0.056 | 0.947±0.011 | 0.770±0.025 | 0.916±0.005 | 0.888±0.014 |
| | SimCLR [16] | 0.882±0.019 | - | 0.767±0.025 | 0.976±0.011 | 0.863±0.036 | 0.876±0.015 | 0.873±0.016 |
| | **IP-IRM (Ours)** | 0.917±0.008 | - | 0.785±0.031 | **0.990**±0.002 | **0.921**±0.009 | 0.916±0.007 | **0.906**±0.011 |
| Shapes3D | VAE [51] | 0.351±0.026 | 0.284±0.009 | **0.820**±0.015 | 0.802±0.054 | 0.421±0.079 | 0.352±0.027 | 0.505±0.028 |
| | $\beta$-VAE [41] | 0.369±0.021 | 0.283±0.012 | 0.782±0.034 | 0.807±0.018 | 0.427±0.025 | 0.368±0.023 | 0.506±0.011 |
| | $\beta$-AnnealVAE [9] | 0.327±0.069 | 0.412±0.049 | 0.743±0.070 | 0.643±0.013 | 0.259±0.021 | 0.328±0.070 | 0.452±0.023 |
| | $\beta$-TCVAE [15] | 0.470±0.035 | 0.291±0.023 | 0.777±0.031 | 0.821±0.054 | 0.439±0.084 | 0.469±0.034 | 0.545±0.032 |
| | Factor-VAE [50] | 0.340±0.021 | 0.316±0.016 | 0.815±0.041 | 0.738±0.043 | 0.319±0.045 | 0.339±0.021 | 0.478±0.020 |
| | SimCLR [16] | 0.535±0.016 | **0.439**±0.030 | 0.678±0.050 | 0.949±0.005 | 0.733±0.055 | 0.536±0.015 | 0.645±0.026 |
| | **IP-IRM (Ours)** | **0.565**±0.023 | 0.420±0.014 | 0.766±0.036 | **0.959**±0.007 | **0.757**±0.025 | **0.565**±0.023 | **0.672**±0.017 |

Table 1: Results on disentanglement metrics of existing unsupervised disentanglement methods, standard SSL (SimCLR [16]) and IP-IRM using CMNIST [2] and Shapes3D [50]. Note that IRS is based on intervening the semantics which requires access to the labels of all the semantics, and hence not applicable for CMNIST dataset. Results are averaged over 4 trails (mean ± std).

**Theorem 1.** *The representation is fully disentangled w.r.t. $\mathcal{G}/\mathcal{D}_{\text{aug}}$ if and only if $\forall c_i \in \mathcal{G}/\mathcal{D}_{\text{aug}}$, the contrastive loss in Eq.* (1) *is invariant to the 2 orbits of partition $\{\mathcal{G}'(c_i \cdot \mathbf{x}), \mathcal{G}'(c_i^{-1} \cdot \mathbf{x})\}$, where $\mathcal{G}' = \mathcal{G}/c_i = \mathcal{D}_{\text{aug}} \times c_1 \times \ldots \times c_{i-1} \times c_{i+1} \times \ldots \times c_k$.*

The maximization in **Step 2** is based on the contra-position of the sufficient condition of Theorem 1. Denote the currently disentangled group as $\mathcal{D}$ (initially $\mathcal{D}_{\text{aug}}$). If we can find a partition $\mathbf{P}^*$ to maximize the loss in Eq. (3), *i.e.*, SSL loss is variant across the orbits, then $\exists h \in \mathcal{G}/\mathcal{D}$ such that the representation of $h$ is entangled, *i.e.*, $\mathbf{P}^* = \{\mathcal{D}(h \cdot \mathbf{x}), \mathcal{D}(h^{-1} \cdot \mathbf{x})\}$. Figure 3 (c) illustrates a discovered partition about color. The minimization in **Step 1** is based on the necessary condition of Theorem 1. Based on the discovered $\mathbf{P}^*$, if we minimize Eq. (2), we can further disentangle $h$ and update $\mathcal{D} \leftarrow \mathcal{D} \times h$. Overall, IP-IRM converges as $\mathcal{G}/\mathcal{D}_{\text{aug}}$ is finite. Note that an improved contrastive objective [92] can further disentangle each $d \in \mathcal{D}_{\text{aug}}$ and achieve full disentanglement *w.r.t.* $\mathcal{G}$.

## 5  Experiments

### 5.1  Unsupervised Disentanglement

**Datasets**. We used two datasets. **CMNIST** [2] has 60,000 digit images with semantic labels of digits (0-9) and colors (red and green). These images differ in other semantics (*e.g.*, slant and font) that are not labeled. Moreover, there is a strong correlation between digits and colors (most 0-4 in red and 5-9 in green), increasing the difficulty to disentangle them. **Shapes3D** [50] contains 480,000 images with 6 labelled semantics, *i.e.*, size, type, azimuth, as well as floor, wall and object color. Note that we only considered the *first three* semantics for evaluation, as the standard augmentations in SSL will contaminate any color-related semantics.

**Settings**. We adopted 6 representative disentanglement metrics: *Disentangle Metric for Informativeness* (DCI) [29], *Interventional Robustness Score* (IRS) [79], *Explicitness Score* (EXP) [72], *Modularity Score* (MOD) [72] and the accuracy of predicting the ground-truth semantic labels by two classification models called *logistic regression* (LR) and *gradient boosted trees* (GBT) [59]. Specifically, DCI and EXP measure the explicitness, *i.e.*, the values of semantics can be decoded from the feature using a linear transformation. MOD and IRS measure the modularity, *i.e.*, whether each feature dimension is equivariant to the shift of a single semantic. See Appendix for more detailed formula of the metrics. In evaluation, we trained CNN-based feature extractor backbones with comparable number of parameters for all the baselines and our IP-IRM. The full implementation details are in Appendix.

**Results**. In Table 1, we compared the proposed IP-IRM to the standard SSL method SimCLR [16] as well as several generative disentanglement methods [51, 41, 9, 15, 50]. On both CMNIST and Shapes3D dataset, IP-IRM outperforms SimCLR regarding all metrics except for only IRS where the most relative gain is 8.8% for MOD. For this MOD, we notice that VAE performs better than

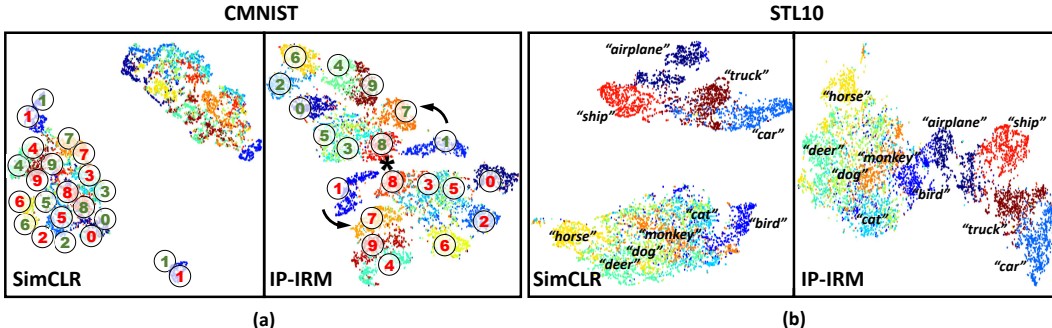

**(a)**

Figure 4: The t-SNE [85] visualizations of learned feature spaces using SimCLR [16] and IP-IRM on CM-NIST [2] and STL10 [22]. For CMIST in (a), we annotate the digit and color near each cluster. We annotate only half of the feature points for SimCLR to avoid clutter. For STL10 in (b), we show the labels of the classes.

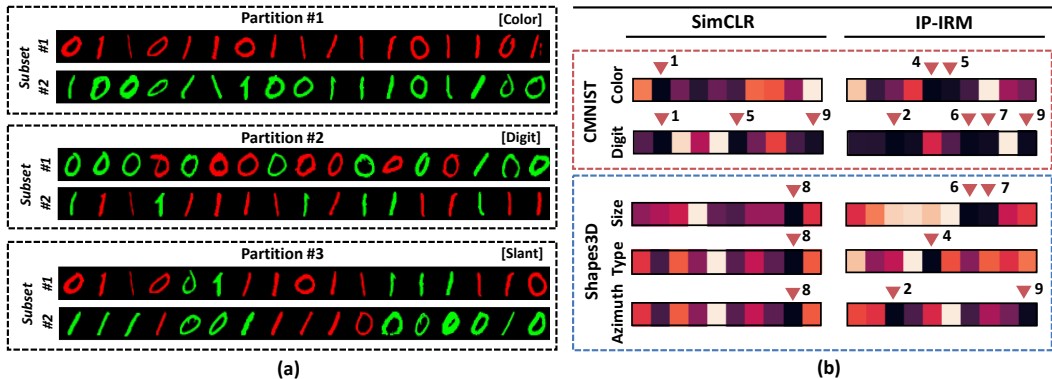

Figure 5: (a) Visualization of the obtained partitions $\mathbf{P}^*$ during training. Each partition has two subset and the displayed images are *randomly* sampled from each subset. (b) Visualization of the variance of each feature dimension when perturbing the semantic indicated on the left. The most equivariant dimensions are indicated by triangles and their corresponding indices.

our IP-IRM by 6 points, *i.e.*, 0.82 v.s. 0.76 for Shapes3D. This is because VAE explicitly pursues a high modularity score through regularizing the dimension-wise independence in the feature space. However, this regularization is adversarial to discriminative objectives [14, 95]. Indeed, we can observe from the column of LR (*i.e.*, the performance of downstream linear classification) that VAE methods have clearly poor performance especially on the more challenging dataset Shapes3D. We can draw the same conclusion from the results of GBT. Different from VAE methods, our IP-IRM is optimized towards disentanglement without such regularization, and is thus able to outperform the others in downstream tasks while obtaining a competitive value of modularity.

**What do IP-IRM features look like?** Figure 4 visualizes the features learned by SimCLR and our IP-IRM on two datasets: CMNIST in Figure 4 (a) and STL10 dataset in Figure 4 (b). In the following, we use Figure 4 (a) as the example, and can easily draw the similar conclusions from Figure 4 (b). On the left-hand side of Figure 4 (a), it is obvious that there is no clear boundary to distinguish the semantic of color in the SimCLR feature space. Besides, the features of the same digit semantic are scattered in two regions. On the right-hand side of (a), we have 3 observations for IP-IRM. 1) The features are well clustered and each cluster corresponds to a specific semantic of either digit or color. This validates the *equivariant* property of IP-IRM representation that it responds to any changes of the existing semantics, *e.g.*, digit and color on this dataset. 2) The feature space has the symmetrical structure for each individual semantic, validating the *decomposable* property of IP-IRM representation. More specifically, i) mirroring a feature (*w.r.t.* "*" in the figure center) indicates the change on the only semantic of color, regardless of the other semantic (digit); and ii) a counterclockwise rotation (denoted by black arrows from same-colored 1 to 7) indicates the change on the only semantic of digit. 3) IP-IRM reveals the true distribution (similarity) of different classes. For example, digits 3, 5, 8 sharing sub-parts (curved bottoms and turnings) have closer feature points in the IP-IRM feature space.

**How does IP-IRM disentangle features?** *1) Discovered* $\mathbf{P}^*$: To visualize the discovered partitions $\mathbf{P}^*$ at each maximization step, we performed an experiment on a binary CMNIST (digit 0 and 1 in color red and green), and show the results in Figure 5 (a). Please kindly refer to Appendix for the full results on CMNIST. First, each partition tells apart a specific semantic into two subsets, *e.g.*, in Partition #1, red and green digits are separated. Second, besides the obvious semantics—digit and color (labelled on the dataset), we can discover new semantics, *e.g.*, the digit slant shown in Partition #3. *2) Disentangled Representation*: In Figure 5 (b), we aim to visualize how equivariant each feature dimension is to the change of each semantic, *i.e.*, a darker color shows that a dimension is more equivariant *w.r.t.* the semantic indicated on the left. We can see that SimCLR fails to learn the decomposable representation, *e.g.*, the 8-th dimension captures azimuth, type and size in Shapes3D. In contrast, our IP-IRM achieves disentanglement by representing the semantics into interpretable dimensions, *e.g.*, the 6-th and 7-th dimensions captures the size, the 4-th for type and the 2-nd and 9-th for azimuth on the Shapes3D. Overall, the results support the justification in Section 4, *i.e.*, we discover a new semantic (affected by $h$) through the partition $\mathbf{P}^*$ at each iteration and IP-IRM eventually converges with a disentangled representation.

## 5.2 Self-Supervised Learning

**Datasets and Settings**. We conducted the SSL evaluations on 2 standard benchmarks following [88, 20, 48]. **Cifar100** [54] contains 60,000 images in 100 classes and **STL10** [22] has 113,000 images in 10 classes. We used SimCLR [16], DCL [20] and HCL [48] as baselines, and learned the representations for 400 and 1000 epochs. We evaluated both linear and $k$-NN ($k = 200$) accuracies for the downstream classification task. Implementation details are in appendix.

| Method | STL10 | | Cifar100 | |
|---|---|---|---|---|
| | $k$-NN | Linear | $k$-NN | Linear |
| *400 epoch training* | | | | |
| SimCLR [16] | 73.60 | 78.89 | 54.94 | 66.63 |
| DCL [20] | 78.82 | 82.56 | 57.29 | 68.59 |
| HCL [48] | 80.06 | 87.60 | 59.61 | 69.22 |
| **SimCLR+IP-IRM** | 79.66 | 84.44 | 59.10 | 69.55 |
| **DCL+IP-IRM** | 81.51 | 85.36 | 58.37 | 68.76 |
| **HCL+IP-IRM** | **84.29** | **87.81** | **60.05** | **69.95** |
| *1,000 epoch training* | | | | |
| SimCLR [16] | 78.60 | 84.24 | 59.45 | 68.73 |
| SimCLR$^\dagger$ [55] | 79.80 | 85.56 | 63.67 | 72.18 |
| **SimCLR$^\dagger$+IP-IRM** | **85.08** | **89.91** | **65.82** | **73.99** |
| Supervised* | - | - | - | 73.72 |
| Supervised*+MixUp [97] | - | - | - | 74.19 |

Table 2: Accuracy (%) of $k$-NN and linear classifiers on STL10 [22] and Cifar100 [54] using the representations of SimCLR [16], DCL [20], HCL [48] and those after incorporating our IP-IRM. SimCLR$^\dagger$ denotes SimCLR with MixUp regularization. Supervised* represents the supervised learning that keeps the same codebase, optimizer and parameters with SSL stage-2 fine-tuning while only adds the learning rate decay at 60 and 80 epoch.

**Results**. We demonstrate our results and compare with baselines in Table 2. Incorporating IP-IRM to the 3 baselines brings consistent performance boosts to downstream classification models in all settings, *e.g.*, improving the linear models by 5.55% on STL10 and 2.92% on Cifar100. In particular, we observe that IP-IRM brings huge performance gain with $k$-NN classifiers, *e.g.*, 4.23% using HCL+IP-IRM on STL10, *i.e.*, the distance metrics in the IP-IRM feature space more faithfully reflects the class semantic differences. This validates that our algorithm further disentangles compared to the standard SSL Moreover, by extending the training process to 1,000 epochs with MixUp [55], SimCLR+IP-IRM achieves further performance boost on both datasets, *e.g.*, 5.28% for $k$-NN and 4.35% for linear classifier over SimCLR baseline on STL10 dataset.

Notably, our SimCLR+IP-IRM surpasses vanilla supervised learning on Cifar100 under the same evaluation setting. Still, the quality of disentanglement cannot be fully evaluated when the training and test samples are identically distributed—while the improved accuracy demonstrates that IP-IRM representation is more equivariant to class semantics, it does not reveal if the representation is decomposable. Hence we present an out-of-distribution (OOD) setting in Section 5.3 to further show this property.

**Is IP-IRM sensitive to the values of hyper-parameters?** 1) $\lambda_1$ *and* $\lambda_2$ *in Eq.* (2) *and Eq.* (3). In Figure 6 (a), we observe that the best performance is achieved with $\lambda_1$ and $\lambda_2$ taking values from 0.2 to 0.5 on both datasets. All accuracies drop sharply if using $\lambda_1 = 1.0$. The reason is that a higher $\lambda_1$ forces the model to push the $\phi$-induced similarity to fixed baseline $\theta = 1$, rather than decrease the loss $\mathcal{L}$ on the pretext task, leading to poor convergence. 2) *The number of epochs*. In Figure 6 (b), we

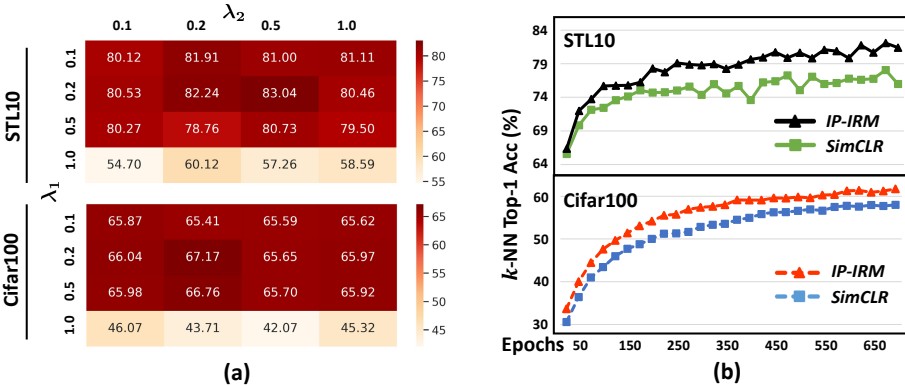

Figure 6: Our ablation study on the STL10 and Cifar100 datasets. (a) The Top-1 accuracy (%) of linear classifiers using different values of $\lambda_1$ and $\lambda_2$ (in Eq. (2) and Eq. (3)), by training for 200 epochs on two datasets. (b) The Top-1 accuracy (%) of $k$-NN classifiers on two datasets, for which we trained the models for 700 epochs and updated $\mathbf{P}$ every 50 epochs.

plot the Top-1 accuracies of using $k$-NN classifiers along the 700-epoch training of two kinds of SSL representations—SimCLR and IP-IRM. It is obvious that IP-IRM converges faster and achieves a higher accuracy than SimCLR. It is worth to highlight that on the STL10, the accuracy of SimCLR starts to oscillate and grow slowly after the 150-th epoch, while ours keeps on improving. This is an empirical evidence that IP-IRM keeps on disentangling more and more semantics in the feature space, and has the potential of improvement through long-term training.

## 5.3 Potential on Large-Scale Data

**Datasets**. We evaluated on the standard benchmark of supervised learning **ImageNet ILSVRC-2012** [25] which has in total 1,331,167 images in 1,000 classes. To further reveal if a representation is decomposable, we used **NICO** [37], which is a real-world image dataset designed for OOD evaluations. It contains 25,000 images in 19 classes, with a strong correlation between the foreground and background in the train split (*e.g.*, most dogs on grass). We also studied the transferability of the learned representation following [30, 52]: FGVC Aircraft (**Aircraft**) [60], Caltech-101 (**Caltech**) [31], Stanford Cars (**Cars**) [93], **Cifar10** [53], **Cifar100** [53], **DTD** [21], Oxford 102 Flowers (**Flowers**) [62], Food-101 (**Food**) [6], Oxford-IIIT Pets (**Pets**) [67] and SUN397 (**SUN**) [91]. These datasets include coarse- to fine-grained classification tasks, and vary in the amount of training data (2,000-75,000 images) and classes (10-397 classes), representing a wide range of transfer learning settings.

**Settings**. For the ImageNet, all the representations were trained for 200 epochs due to limited computing resources. We followed the common setting [80, 36], using a linear classifier, and report Top-1 classification accuracies. For NICO, we fixed the ImageNet pre-trained ResNet-50 backbone and fine-tuned the classifier. See appendix for more training details. For the transfer learning, we followed [30, 52] to report the classification accuracies on Cars, Cifar-10, Cifar-100, DTD, Food, SUN and the average per-class accuracies on Aircraft, Caltech, Flowers, Pets. We call them uniformly as Accuracy. We used the few-shot $n$-way-$k$-shot setting for model evaluation. Specifically, we randomly sampled 2,000 episodes from the *test* splits of above datasets. An episode contains $n$ classes, each with $k$ training samples and 15 testing samples, where we fine-tuned the linear classifier (backbone weights frozen) for 100 epochs on the training samples, and evaluated the classifier on the testing samples. We evaluated with $n = k = 5$ (results of $n = 5, k = 20$ in Appendix).

**ImageNet and NICO**. In Table 3 ImageNet accuracy, our IP-IRM achieves the best performance over all baseline models. Yet we believe that this does not show the full potential of IP-IRM, because ImageNet is a larger-scale dataset with many semantics, and it is hard to achieve a full disentanglement of all semantics within the limited 200 epochs. To evaluate the feature decomposability of IP-IRM, we compared the performance on NICO with various SSL baselines in Table 3, where our approach significantly outperforms the baselines by 1.5-4.2%. This validates IP-IRM feature is more decomposable—if each semantic feature (*e.g.*, background) is decomposed in some fixed dimensions and some classes vary with such semantic, then the classifier will recognize this as

| Method | ImageNet | NICO |
|---|---|---|
| InsDis [90] | 56.5 | 65.6 |
| PCL [56] | 61.5 | 72.6 |
| PIRL [61] | 63.6 | 69.1 |
| MoCo-v1 [36] | 60.6 | 69.3 |
| SimCLR (repro.) [16] | 63.1 | 64.5 |
| MoCo-v2 (repro.) [18] | 67.3 | 78.0 |
| SimSiam (repro.) [19] | 68.8 | 66.7 |
| **SimCLR+IP-IRM** | 64.8 | 66.7 |
| **MoCo-v2+IP-IRM** | 67.6 | **79.5** |
| **SimSiam+IP-IRM** | **69.1** | 70.9 |

Table 3: ImageNet and NICO Top-1 Accuracy (%) of linear classifiers trained on the representations learnt with different SSL methods.

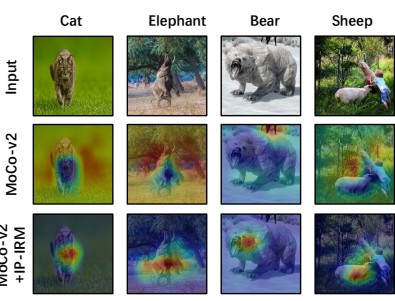

Figure 7: Visualization of CAM [98] on images from NICO [37] dataset using representations of the baseline MoCo-v2 [18] and our IP-IRM.

| Method | Aircraft | Caltech | Cars | Cifar10 | Cifar100 | DTD | Flowers | Food | Pets | SUN | Average |
|---|---|---|---|---|---|---|---|---|---|---|---|
| InsDis [90] | 35.07 | 75.97 | 37.49 | 51.49 | 57.61 | 69.38 | 77.35 | 50.01 | 66.38 | 74.97 | 59.57 |
| PCL [56] | **36.86** | 90.72 | 39.68 | 59.26 | 60.78 | 69.53 | 67.50 | 57.06 | **88.31** | 84.51 | 65.42 |
| PIRL [61] | 36.70 | 78.63 | 39.21 | 49.85 | 55.23 | 70.43 | 78.37 | 51.61 | 69.40 | 76.64 | 60.61 |
| MoCo-v1 [36] | 35.31 | 79.60 | 36.35 | 46.96 | 51.62 | 68.76 | 75.42 | 49.77 | 68.32 | 74.77 | 58.69 |
| MoCo-v2 [18] | 31.98 | 92.32 | 41.47 | 56.50 | 63.33 | 78.00 | 80.05 | 57.25 | 83.23 | 88.10 | 67.22 |
| **IP-IRM (Ours)** | 32.98 | **93.16** | **42.87** | **60.73** | **68.54** | **79.30** | **82.68** | **59.61** | 85.23 | **89.38** | **69.44** |

Table 4: Accuracy (%) of 5-way-5-shot few-shot evaluation using the image representation learned on ImageNet [25]. More detailed results are given in Appendix.

a non-discriminative variant feature and hence focus on other more discriminative features (*i.e.*, foreground). In this way, even though some classes are confounded by those non-discriminative features (*e.g.*, most of the "dog" images are with "grass" background), the fixed dimensions still help classifiers neglect those non-discriminative ones. We further visualized the CAM [98] on NICO in Figure 7, which indeed shows that IP-IRM helps the classifier focus on the foreground regions.

**Few-Shot Tasks.** As shown in Table 4, our IP-IRM significantly improves the performance of 5-way-5-shot setting, *e.g.*, we outperform the baseline MoCo-v2 by 2.2%. This is because IP-IRM can further disentangled $\mathcal{G} \setminus \mathcal{D}_{\text{aug}}$ over SSL, which is essential for representations to generalize to different downstream class domains (recall Corollary 2 of Lemma 1). This is also in line with recent works [86] showing that a disentangled representation is especially beneficial in low-shot scenarios, and further demonstrates the importance of disentanglement in downstream tasks.

## 6  Conclusion

We presented an unsupervised disentangled representation learning method called Iterative Partition-based Invariant Risk Minimization (IP-IRM), based on Self-Supervised Learning (SSL). IP-IRM iteratively partitions the dataset into semantic-related subsets, and learns a representation invariant across the subsets using SSL with an IRM loss. We show that with theoretical guarantee, IP-IRM converges with a disentangled representation under the group-theoretical view, which fundamentally surpasses the capabilities of existing SSL and fully-supervised learning. Our proposed theory is backed by strong empirical results in disentanglement metrics, SSL classification accuracy and transfer performance. IP-IRM achieves disentanglement without using generative models, making it widely applicable on large-scale visual tasks. As future directions, we will continue to explore the application of group theory in representation learning and seek additional forms of inductive bias for faster convergence.

## Acknowledgments and Disclosure of Funding

The authors would like to thank all reviewers for their constructive suggestions. This research is partly supported by the Alibaba-NTU Joint Research Institute, the A*STAR under its AME YIRG Grant (Project No. A20E6c0101), and the Singapore Ministry of Education (MOE) Academic Research Fund (AcRF) Tier 2 grant.

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
