# Appendix

This is the Appendix for "Self-Supervised Learning Disentangled Group Representation as Feature". Table .1 summarizes the abbreviations and the symbols used in the main paper.

| Abbreviation/Symbol | Meaning |
|---|---|
| *Abbreviation* | |
| SSL | Self-supervised Learning |
| SL | Supervised Learning |
| DCI | Disentangle Metric for Informativeness |
| IRS | Interventional Robustness Score |
| EXP | Explicitness Score |
| MOD | Modularity Score |
| LR | Logistic Regression |
| GBT | Gradient Boosted Trees |
| OOD | Out-Of-Distributed |
| *Symbol in Theory* | |
| $\mathcal{U}$ | Semantic space |
| $\mathcal{X}$ | Vector space |
| $\mathcal{I}$ | Image space |
| $\mathcal{G}$ | Group |
| $\mathcal{G}(\mathbf{x})$ | Group orbit *w.r.t.* $\mathcal{G}$ containing the sample $\mathbf{x}$ |
| $\varphi$ | Image generation process $\mathcal{U} \to \mathcal{I}$ |
| $\phi$ | Visual representation $\mathcal{I} \to \mathcal{X}$ |
| $f$ | Semantic representation $\mathcal{U} \to \mathcal{X}$ |
| $m$ | The number of decomposed subgroups |
| *Symbol in Algorithm* | |
| $\mathbf{P}$ | Partition of dataset |
| $\mathbf{P}^*$ | Learned partition through Eq. (3) |
| $\mathcal{P}$ | Set of partitions used in Eq. (2) |
| $N$ | Number of training images |
| $\theta$ | "Dummy" parameter used by IRM |
| $I$ | Image |
| $\mathcal{X}_k$ | The set of features in the subset $k$ |
| $\mathcal{X}^*$ | The set of augmented view features |
| $\lambda_1, \lambda_2$ | Hyper-parameters in Eq. (2) and (3) |

Table .1: List of abbreviations and symbols used in the paper.

This appendix is organized as follows:

- Section A provides the preliminary knowledge about the group theory. We also provide a formal definition of "fixed" and "affected" used in Definition 1.

- Section B gives the proofs of Definition 2, Lemma 1 and Theorem 1.

- Section C shows the implementation details of our experiments in Section 5 of our main paper.

- Section D presents the additional experimental results.

# A  Preliminaries

A group is a set together with a binary operation, which takes two elements in the group and maps them to another element. For example, the set of integers is a group under the binary operation of plus. We formalize the notion through the following definition.

**Binary Operation**. A binary operation $\cdot$ on a set $\mathcal{S}$ is a function mapping $\mathcal{S} \times \mathcal{S}$ into $\mathcal{S}$. For each $(s_1, s_2) \in \mathcal{S} \times \mathcal{S}$, we denote the element $\cdot(s_1, s_2)$ by $s_1 \cdot s_2$.

**Group**. A group $\langle \mathcal{G}, \cdot \rangle$ is a set $\mathcal{G}$, closed under a binary operation $\cdot$, such that the following axioms hold:

1. *Associativity.* $\forall g_1, g_2, g_3 \in \mathcal{G}$, we have $(g_1 \cdot g_2) \cdot g_3 = g_1 \cdot (g_2 \cdot g_3)$.
2. *Identity Element.* $\exists e \in \mathcal{G}$, such that $\forall g \in \mathcal{G}$, $e \cdot g = g \cdot e = g$.
3. *Inverse.* $\forall g \in \mathcal{G}, \exists g' \in \mathcal{G}$, such that $g \cdot g' = g' \cdot g = e$.

Groups often arise as transformations of some space, such as a set, vector space, or topological space. Consider an equilateral triangle. The set of clockwise rotations *w.r.t.* its centroid to retain its appearance forms a group $\{60°, 120°, 180°\}$, with the last element corresponding to an identity mapping. We say this group of rotations act on the triangle, which is formally defined below.

**Group Action**. Let $\mathcal{G}$ be a group with binary operation $\cdot$ and $\mathcal{S}$ be a set. An action of $\mathcal{G}$ on $\mathcal{S}$ is a map $\pi : \mathcal{G} \to \mathrm{Hom}(\mathcal{S}, \mathcal{S})$ so that $\pi(e) = \mathrm{id}_{\mathcal{S}}$ and $\pi(g) \circ \pi(h) = \pi(g \cdot h)$, where $g, h \in \mathcal{G}$, $\circ$ denotes functional composition. $\forall g \in \mathcal{G}, s \in \mathcal{S}$, denote $\pi(g)(s)$ as $g \circ s$.

In our formulation, we have a group $\mathcal{G}$ acting on the semantic space $\mathcal{U}$. For example, consider the color semantic, which can be mapped to a circle representing the hue. Hence the group acting on it corresponds to rotations, similar to the triangle example, *e.g.*, $\pi(g)$ may correspond to rotating a color in $\mathcal{S}$ clockwise by $30°$. In the context of representation learning, we are interested to learn a feature space to reflect $\mathcal{G}$, formally defined below.

**Group Representation**. Let $\mathcal{G}$ be a group. A representation of $\mathcal{G}$ (or $\mathcal{G}$-representation) is a pair $(\pi, \mathcal{X})$, where $\mathcal{X}$ is a vector space and $\pi : \mathcal{G} \to \mathrm{Hom}_{vect}(\mathcal{X}, \mathcal{X})$ is a group action, *i.e.*, for each $g \in \mathcal{G}$, $\pi(g) : \mathcal{X} \to \mathcal{X}$ is a linear map.

Intuitively, each $g \in \mathcal{G}$ corresponds to a linear map, *i.e.*, a matrix $\mathbf{M}_g$ that transforms a vector $\mathbf{x} \in \mathcal{X}$ to $\mathbf{M}_g \mathbf{x} \in \mathcal{X}$. Finally, there is a decomposition of semantic space and the group acting on it in our definition of disentangled representation. The decomposition of semantic space is based on the Cartesian product $\times$. A similar concept is defined *w.r.t.* group.

**Direct Product of Group**. Let $\mathcal{G}_1, \ldots, \mathcal{G}_n$ be groups with the binary operation $\cdot$. Let $a_i, b_i \in \mathcal{G}_i$ for $i \in \{1, \ldots, n\}$. Define $(a_1, \ldots, a_n) \cdot (b_1, \ldots, b_n)$ to be the element $(a_1 \cdot b_1, \ldots, a_n \cdot b_n)$. Then $\mathcal{G}_1 \times \ldots \times \mathcal{G}_n$ or $\prod_{i=1}^n \mathcal{G}_i$ is the direct product of the groups $\mathcal{G}_1, \ldots, \mathcal{G}_n$ under the binary operation $\cdot$.

With this, we can formally define $\mathcal{X}_i, i \in \{1, \ldots, m\}$ *is only affected by the action of $\mathcal{G}_i$ and fixed by the action of other subgroups*: $(\pi | \mathcal{G}_j, \mathcal{X}_i)_{j \neq i}$ is a trivial sub-representation ("fixed"), *i.e.*, for each $g \in \mathcal{G}_j, j \neq i$, $\pi(g)$ is the identity mapping $\mathrm{id}_{\mathcal{X}_i}$, and $(\pi | \mathcal{G}_j, \mathcal{X}_i)_i$ is non-trivial ("affected").

# B  Proof

## B.1  Proof of Definition 2

$\mathcal{D}$ **Defines a Partition of $\mathcal{X}$**. We will show that $\mathcal{D}$ defines an equivalence relation on $\mathcal{X}$, which naturally leads to a partition of $\mathcal{X}$. For $\mathbf{x}_1, \mathbf{x}_2 \in \mathcal{X}$, let $\mathbf{x}_1 \sim \mathbf{x}_2$ if and only if $\exists g \in \mathcal{D}$ such that $g \cdot \mathbf{x}_1 = \mathbf{x}_2$. We show that $\sim$ satisfies the three properties of equivalence relation. 1) Reflexive: $\forall \mathbf{x} \in \mathcal{X}$, we have $e \cdot \mathbf{x} = \mathbf{x}$, hence $\mathbf{x} \sim \mathbf{x}$. 2) Symmetric: Suppose $\mathbf{x}_1 \sim \mathbf{x}_2$, *i.e.*, $g \cdot \mathbf{x}_1 = \mathbf{x}_2$ for some $g \in \mathcal{D}$. Then $g^{-1} \cdot \mathbf{x}_2 = \mathbf{x}_1$, *i.e.*, $\mathbf{x}_2 \sim \mathbf{x}_1$. 3) Transitive: if $\mathbf{x}_1 \sim \mathbf{x}_2$ and $\mathbf{x}_2 \sim \mathbf{x}_3$, then $g_1 \cdot \mathbf{x}_1 = \mathbf{x}_2$ and $g_2 \cdot \mathbf{x}_2 = \mathbf{x}_3$ for some $g_1, g_2 \in \mathcal{D}$. Hence $(g_2 g_1) \cdot \mathbf{x}_1 = \mathbf{x}_3$ and $\mathbf{x}_1 \sim \mathbf{x}_3$.

**Number of Orbits**. Recall that $\mathcal{G}$ acts transitively on $\mathcal{X}$ (see Section 4). We consider the non-trivial case where the action of $\mathcal{G}$ is faithful, *i.e.*, the only group element that maps all $\mathbf{x} \in \mathcal{X}$ to itself is the identity element $e$. Let $\mathcal{K} = \mathcal{G} / \mathcal{D} = g_1 \times \ldots \times g_k$. We will show that each $c \in \mathcal{K}$ corresponds to a unique orbit. 1) $\forall c \neq e \in \mathcal{K}$, $\mathcal{D}(\mathbf{x}) \neq \mathcal{D}(c \cdot \mathbf{x})$. Suppose $\exists c \neq e \in \mathcal{K}$, such that for some

$\mathbf{x} \in \mathcal{X}$, the action of $c$ on each $\mathbf{x}_1 \in \mathcal{D}(\mathbf{x})$ corresponds to the identity mapping. One can show that for every different orbit, *i.e.*, $\mathcal{D}(c' \cdot \mathbf{x}) \neq \mathcal{D}(\mathbf{x})$, the action of $c$ on each $\mathbf{x}_2 \in \mathcal{D}(\mathbf{x})$ is also identity mapping. As $\mathcal{D}$ partitions $\mathcal{X}$ into orbits *w.r.t.* $\mathcal{D}$, this means that the action of $c$ is identity mapping on all $\mathbf{x} \in \mathcal{X}$, which contradicts with the action of $\mathcal{G}$ being faithful. 2) The previous step shows that non-identity group elements in $\mathcal{K}$ lead to a different orbit. We need to further show that these orbits are unique, *i.e.*, $\forall c, c' \neq e \in \mathcal{K}$, if $c \neq c'$, then $\mathcal{D}(c \cdot \mathbf{x}) \neq \mathcal{D}(c' \cdot \mathbf{x})$. Suppose $c' \cdot \mathbf{x} = c \cdot \mathbf{x}$, *i.e.*, $c^{-1}c' \cdot \mathbf{x} = c^{-1}c \cdot \mathbf{x} = \mathbf{x}$, so $c^{-1}c' \in \mathcal{G}_{\mathbf{x}}$, where $\mathcal{G}_{\mathbf{x}}$ is the point stabilizer of $\mathbf{x}$. As the action of $\mathcal{G}$ is faithful, $\mathcal{G}_{\mathbf{x}} = \{e\}$. Hence $c \cdot \mathbf{x} = c' \cdot \mathbf{x}$ implies $c = c'$.

## B.2 Details of Lemma 1

We will first prove Lemma 1 by showing the representation is $\mathcal{G}$-equivariant, followed by showing that $\mathcal{D}$ and $\mathcal{G}/\mathcal{D}$ are decomposable and finally showing that $\prod_{d \in \mathcal{D}} d$ is not decomposable. We will then present more details on the 4 corollaries.

**Proof of $\mathcal{G}$-equivariant.** Suppose that the training loss $-\log \frac{\exp(\mathbf{x}_i^T \mathbf{x}_j)}{\sum_{\mathbf{x} \in \mathcal{X}} \exp(\mathbf{x}_j^T \mathbf{x})}$ is minimized, yet $\exists \mathbf{x}_a = \mathbf{x}_b \in \mathcal{X}$ for $a \neq b$. Let $\mathbf{x}_i \in \mathcal{X}$ in the denominator, and we have $\mathbf{x}_j^T \mathbf{x}_i = \cos(\theta_{i,j}) \|\mathbf{x}_i\| \|\mathbf{x}_j\|$, where $\theta_{i,j}$ is the angle between the two vectors. When $\mathbf{x}_i = \mathbf{x}_j$, $\cos(\theta_{i,j}) = 1$. So keeping $\|\mathbf{x}_i\| \|\mathbf{x}_j\|$ constant (*i.e.*, the same regularization penalty such as L2), $\mathbf{x}_j^T \mathbf{x}_i$ can be further reduced if $\mathbf{x}_i \neq \mathbf{x}_j$, which reduces the training loss. This contradicts with the earlier assumption. Hence by minimizing the training loss, we can achieve sample-equivariant, *i.e.*, different samples have different features. Note that this does not necessarily mean group-equivariant. However, the variation of training samples is all we know about the group action of $\mathcal{G}$, and we establish that the action of $\mathcal{G}$ is transitive on $\mathcal{X}$, hence we use the sample-equivariant features as the approximation of $\mathcal{G}$-equivariant features.

**Proof of Decomposability between $\mathcal{D}$ and $\mathcal{G}/\mathcal{D}$.** Recall the semantic representation $f : \mathcal{U} \to \mathcal{I} \to \mathcal{X}$, which is show to be $\mathcal{G}$-equivariant in the previous step. Consider a non-decomposable representation where $\mathcal{X}$ is affected by the action of both $\mathcal{D}$ and $\mathcal{G}/\mathcal{D}$. Let $\mathcal{X} = \mathcal{X}_d \times \mathcal{X}_c$, where both sub-spaces are affected by the action of the two groups. In particular, denote the semantic representation $f_c : (\mathcal{U}_d \times \mathcal{U}_c) \to \mathcal{X}_c$, where $\mathcal{U}_d$ is affected by the action of $\mathcal{D}$ (recall that $\mathcal{G}$ affects $\mathcal{U}$ and $\mathcal{X}$ through the equivariant map in Figure 1) and $\mathcal{U}_c$ is affected by the action of $\mathcal{G}/\mathcal{D}$. From here, we will construct a representation where $\mathcal{X}_c$ is only affected by the action of $\mathcal{G}/\mathcal{D}$ with a lower training loss.

Specifically, we aim to assign a $d_i^* \in \mathcal{U}_d, i \in \{1, \ldots, k\}$ to the $i$-th orbit, which is given by:

$$d_1^*, \ldots, d_k^* = \arg\min_{(d_1, \ldots, d_k)} \mathbb{E}_{i,j \in \{1, \ldots, k\}} f_c(d_i, c_i)^T f_c(d_j, c_j), \tag{B.1}$$

where $c_i \in \mathcal{U}_c$ is the value of $U_c$ for $i$-th orbit. Now define $f_c^* : \mathcal{U}_c \to \mathcal{X}_c$ given by $f_c^*(c_i) = f_c(d_i^*, c_i) \forall i \in \{1, \ldots, k\}$. Using this new $f_c^*$ has two outcomes:

1) $\mathbf{x}_i^T \mathbf{x}_j$ in the numerator is the linear combination of the dot similarity induced from $\mathcal{X}_d$ and $\mathcal{X}_c$. And the dot similarity induced from $\mathcal{X}_c$ is increased, as inside each orbit, the value in $\mathcal{X}_c$ is the same (maximized similarity);

2) The denominator is now reduced. This is because the denominator is proportional to $\mathbb{E}_{i,j \in \{1, \ldots, k\}} \mathbb{E}_{d,d' \in \mathcal{U}_d} f_c(d, c_i)^T f_c(d', c_j)$, and we have already selected the best set $d_i^*$ that minimizes the expected dot similarities across orbits.

As the in-orbit dot similarity increases (numerator), and the cross-orbit dot similarity decreases (denominator), the training loss is reduced by decomposing a separate sub-space $\mathcal{X}_c$ affected only by the action of $\mathcal{G}/\mathcal{D}$ with $f_c^*$. Furthermore, note that a linear projector is used in SSL to project the features into lower dimensions, and a linear weight is used in SL. To isolate the effect of $\mathcal{D}$ to maximize the similarity of in-orbit samples (numerator) and exploit the action of $\mathcal{G}/\mathcal{D}$ to minimize the similarity of cross-orbit samples (denominator), the effect of $\mathcal{D}$ and $\mathcal{G}/\mathcal{D}$ on $\mathcal{X}_d$ must be separable by a linear layer, *i.e.*, decomposable. Combined with the earlier proof that $\mathcal{X}_c$ is only affected by the action of $\mathcal{G}/\mathcal{D}$, without loss of generality, we have the decomposition $\mathcal{X} = \mathcal{X}_d \times \mathcal{X}_c$ affected by $\mathcal{D}$ and $\mathcal{G}/\mathcal{D}$, respectively.

**Proof of Non-Decomposability of $d \in \mathcal{D}$.** We will show that for a representation with $d \in \mathcal{D}$ decomposed, there exists a non-decomposable representation that achieves the same expected dot similarity, hence having the same contrastive loss. Without loss of generality, consider $d_1, d_2 \in \mathcal{D}$

acting on the semantic attribute space $\mathcal{U}_1, \mathcal{U}_2$, respectively. Let $f$ be a decomposable representation such that there exists feature subspaces $\mathcal{X}_1, \mathcal{X}_2 \in \mathcal{X}$ affected only by the action of $d_1, d_2$, respectively. Denote $f_1 : \mathcal{U}_1 \to \mathcal{X}_1, f_2 : \mathcal{U}_2 \to \mathcal{X}_2$. Now we define a non-decomposable representation with mapping $f_1' : (\mathcal{U}_1, \mathcal{U}_2) \to \mathcal{X}_1$ and $f_2' : (\mathcal{U}_1, \mathcal{U}_2) \to \mathcal{X}_2$, given by $f_1'(U_1 = u_1, U_2 = u_2) = \frac{1}{\sqrt{2}}(f_1(U_1 = u_1) + f_2(U_2 = u_2))$ and $f_2'(U_1 = u_1, U_2 = u_2) = \frac{1}{\sqrt{2}}(f_1(U_1 = u_1) - f_2(U_2 = u_2))$. Now for any pair of samples with semantic $(u_1, u_2)$ and $(u_1', u_2')$, $u_1, u_1' \in \mathcal{U}_1$ and $u_2, u_2' \in \mathcal{U}_2$, the dot similarity induced from the subspace $\mathcal{X}_1, \mathcal{X}_2$ is given by $f_1(u_1)^2 + f_2(u_2)^2$. Therefore, the decomposed and non-decomposed representations yield the same expected dot similarity over all pairs of samples, and have the same contrastive loss.

**Corollary 1**. This follows immediately from the proof on $\mathcal{G}$-equivariance.

**Corollary 2**. The same proof above holds for the SL case with $\mathbf{x}_i \in \mathcal{X}$, where $\mathcal{X}$ is the set of classifier weights. In this view, each sample in the class can be seen as an augmented view (augmented by shared attributes such as view angle, pose, etc) of the class prototype. In downstream learning, the *shared* attributes are not discriminative, hence the performance is affected mostly by $\mathcal{G}/\mathcal{D}$. For example, if the groups corresponding to "species" and "shape" act on the same feature subspace (entangled), such that "species"="bird" always have "shape"="streamlined" feature, this representation does not generalize to downstream tasks of classifying birds without streamlined shape (*e.g.*, "kiwi").

**Corollary 3**. In SL and SSL, the model essentially receives supervision on attributes that are not discriminative towards downstream tasks, through augmentations and in-class variations, respectively. The group $\mathcal{D}$ acts on the semantic space of these attributes, hence $|\mathcal{D}|$ determines the amount of supervision received. With a large $|\mathcal{D}|$, the model filters off more irrelevant semantics and $\mathcal{G}/\mathcal{D}$ more accurately describe the differences between classes. Note that the standard image augmentations in SSL are also used in SL, making $|\mathcal{D}|$ even larger in SL.

**Corollary 4**. When the number of samples in some orbit(s) is smaller than $|\mathcal{D}(\mathbf{x})|$, this has two consequences that prevent disentanglement: 1) The $\mathcal{G}$-equivariance is not guaranteed as the training samples do not fully describe $\mathcal{G}$. 2) The decomposability is not guaranteed as the decomposed $f_c^*, f_d^*$ in the previous proof only generalizes to the seen combination of the value in $\mathcal{U}_d \times \mathcal{U}_c$.

### B.3  Proof of Theorem 1

We will first revisit the Invariant Risk Minimization (IRM). Let $\mathcal{I}$ be the image space, $\mathcal{X}$ the feature space, $\mathcal{Y}$ the classification output space (*e.g.*, the set of all probabilities of belonging to each class), the feature extractor backbone $\phi : \mathcal{I} \to \mathcal{X}$ and the classifier $\omega : \mathcal{X} \to \mathcal{Y}$. Let $\mathcal{E}_{tr}$ be a set of training environments, where each $e \in \mathcal{E}_{tr}$ is a set of images. IRM aims to solve the following optimization problem:

$$\min_{\phi, \omega} \sum_{e \in \mathcal{E}_{tr}} R^e(\omega \circ \phi)$$
$$\text{subject to } \omega \in \arg\min_{\bar{\omega}} R^e(\bar{\omega} \circ \phi) \ \forall e \in \mathcal{E}_{tr}, \tag{B.2}$$

where $R^e(\omega \circ \phi)$ is the empirical classification risk in the environment $e$ using backbone $\phi$ and classifier $\omega$. Conceptually, IRM aims to find a representation $\phi$ such that the optimal classifier on top of $\phi$ is the same for all environments. As Eq. (B.2) is a challenging, bi-leveled optimization problem, it is initiated into the practical version:

$$\min_{\phi} \sum_{e \in \mathcal{E}_{tr}} R^e(\phi) + \lambda \|\nabla_{\omega=1.0} R^e(\omega \cdot \phi)\|^2, \tag{B.3}$$

where $\lambda$ is the regularizer balancing between the ERM term and invariant term.

The above IRM is formulated for supervised training. In SSL, there is no classifier mapping from $\mathcal{X} \to \mathcal{Y}$. Instead, there is a projector network $\sigma : \mathcal{X} \to \mathcal{Z}$ mapping features to another feature space $\mathcal{Z}$, and Eq. (1) is used to compute the similarity with positive key (numerator) and negative keys (denominator) in $\mathcal{Z}$. Note that $h$ in SSL is not equivalent to $\phi$ in SL, as $\sigma$ itself does not generate the probability output like $\omega$, rather, the comparison between positive and negative keys does.

In fact, the formulation of contrastive IRM is given by Corollary 2 of Lemma 1, which says that SL is a special case of contrastive learning, and the set of all classifier weights is the positive and negative

key space. In IRM with SL, we are trying to find a set of weights $\omega_{\text{SL}}$ from the classifier weights space (*e.g.*, $\mathbb{R}^{d \times c}$ with feature dimension as $d$ and number of classes as $c$) that achieves invariant prediction. Hence in IRM with SSL, we are trying to find a set of keys $\omega_{\text{SSL}}$ from the key space (*e.g.*, $\mathbb{R}^{z \times n}$ with $z$ being the dimension of $\mathcal{Z}$ and number of positive and negative keys as $n$) that achieves invariant prediction by differentiating a sample with negative keys (Note that the similarity with positive keys is maximized and fixed using standard SSL training by decomposing augmentations and other semantics as in Lemma 1). Specifically, in IP-IRM, the 2 subsets in each partition form the set of training environments $\mathcal{E}_{tr}$.

**Proof of the Sufficient Condition**. Suppose that the representation is fully disentangled *w.r.t.* $\mathcal{G}/\mathcal{D}_{\text{aug}}$. By Definition 1, there exists subspace $\mathcal{X}_i \in \mathcal{X}$ affected only by the action of $c_i \in \mathcal{G}/\mathcal{D}_{\text{aug}}$. For each partition given by $\{\mathcal{G}'(c_i \cdot \mathbf{x}), \mathcal{G}'(c_i^{-1} \cdot \mathbf{x})\}$, let the projector network $\sigma^* : \mathcal{X}' \to \mathcal{Z}$, where $\mathcal{X}' = \mathcal{X}_1 \times \ldots \times \mathcal{X}_{i-1} \times \mathcal{X}_{i+1} \times \ldots \times \mathcal{X}_k$. Note that $\sigma^*$ can be achieved in the parameter space of a linear layer, as the subspace $\mathcal{X}_i$ is decomposed into fixed dimensions for all samples, which can be filtered out by a linear layer by setting weights associated to those dimensions as $0$. Moreover, the resulting space $\mathcal{Z}$ (*i.e.*, space of positive and negative keys) is affected only by the action of $\mathcal{G}'$. As the in-orbit group corresponds to $\mathcal{G}'$, the values in $\mathcal{X}_i$ are not discriminative towards SSL objective. Hence there exists $\sigma^*$ mapping to $\mathcal{Z}$ that is optimal in both orbits, *i.e.*, minimizing the contrastive IRM loss.

**Proof of the Necessary Condition**. Suppose that the contrastive IRM loss is minimized for the partition $\{\mathcal{G}'(c_i \cdot \mathbf{x}), \mathcal{G}'(c_i^{-1} \cdot \mathbf{x})\}$. We will show that $c_i$ is disentangled. First we will consider the space $\mathcal{Z}$ (mapped from $\mathcal{X}$ by the projector network). If $\mathcal{Z}$ is affected by the action of $c_i$ and $\mathcal{G}'$, let $\omega_1^* \in \arg\min_{\bar{\omega}} R^1(\bar{\omega} \circ \phi)$ that optimizes the contrastive loss in the first orbit (by exploiting the equivariance of $\mathcal{G}'$). Given a fixed $\phi$, $\omega_1^*$ is unique as the contrastive loss is convex. The IRM constraint requires that $\omega_1^* = \omega_2^*$, which means that the action of $c_i$ on $\mathcal{Z}$ corresponds to identity mapping. Yet this contradicts with the equivariant property from Corollary 1 of Lemma 1. Therefore when the contrastive IRM loss is minimized, $\mathcal{Z}$ cannot be affected by the action of $c_i$. Given the linear projector network, there are two possibilities for $\mathcal{X}$: 1) $\mathcal{X} = \mathcal{X}_i' \times \mathcal{X}'$ where $\mathcal{X}_i'$ is affected by the action of $c_i$ and $\mathcal{G}'$, and $\mathcal{X}'$ is affected by the action of $\mathcal{G}'$. In this way, the projector $\sigma$ can discard $\mathcal{X}_i'$ to obtain $\mathcal{Z}$ unaffected by $c_i$. However, under the representation $\phi$ leading to the feature space with decomposition $\mathcal{X} = \mathcal{X}_i' \times \mathcal{X}'$, the optimal $\omega$ in each orbit will exploit $\mathcal{X}_i'$, which is discriminative as it is affected by $\mathcal{G}'$. Therefore, there exists no $\omega$ that is *simultaneously* optimal between the two orbits under this representation. 2) $\mathcal{X} = \mathcal{X}_i \times \mathcal{X}'$ where $\mathcal{X}_i$ is affected only by the action of $c_i$, and $\mathcal{X}'$ is affected by the action of $\mathcal{G}'$, *i.e.*, the representation is disentangled with $c_i$. In this way, we have $\omega$ that is simultaneously optimal across orbits as shown in the proof of the sufficient condition.

**Example of Step 2**. As we consider two orbits in each partition corresponding to $g_i$, we denote $\mathcal{U}_i \in \{0, 1\} \forall i \in \{1, \ldots, k\}$ as a binary attribute space affected by $g_i$. We will show how maximizing the contrastive IRM loss will lead to the partition where cross-orbit group element $h \in \mathcal{G}/\mathcal{D}$.

From Lemma 1, the representation $f$ is equivariant under the action of $\mathcal{G}$, *i.e.*, the SSL loss $\mathcal{L}$ reveal the information about $g \in \mathcal{G}$. Specifically, each pair of samples in a subset corresponds to a group element whose action transform the attribute of the first sample to the second sample. In a subset, if more group elements corresponding to pair-wise transformation are identity mapping, the samples in the group are more similar in semantics. With the equivariant property of the representation, the features in the subset are also more similar, leading to larger $\mathcal{L}$ (more difficult to distinguish two samples apart).

Given binary semantic attributes, the similarity in the semantics is measured by the hamming distance, *i.e.*, for $u_1, u_2 \in \prod_{i=1}^k \mathcal{U}_i$, their hamming distance $d_H(u_1, u_2)$ is given by the number of different bits (*e.g.*, $d_H(u_1 = 01, u_2 = 11) = 1$). Denote the set of semantic attributes of the samples in the two orbits in space $\mathcal{U}_1, \ldots, \mathcal{U}_k$ as $\mathcal{S}_1$ and $\mathcal{S}_2$, respectively. From the equivariant property of Lemma 1, the first term in Eq. (3) $\mathcal{L}(\phi, \theta = 1.0, k = 1, \mathbf{P}) + \mathcal{L}(\phi, \theta = 1.0, k = 2, \mathbf{P})$ is maximized when the average hamming distance in the two orbits $d(\mathcal{S}_1) + d(\mathcal{S}_2)$ is minimized, where $d(\mathcal{S})$ is given by:

$$d(\mathcal{S}) = \frac{1}{|\mathcal{S}|^2} \sum_{u \in \mathcal{S}} \sum_{u' \in \mathcal{S}} d_H(u, u'). \tag{B.4}$$

Without loss of generality, an arbitrary data partition is illustrated in Figure B.1 (a), where we arrange the order of the samples in each orbit, such that those with $U_t = 0, t \in \{1, \ldots, k\}$ come first in the

subset $\mathcal{S}_1$, and those with $U_t = 1$ come first in the subset $\mathcal{S}_2$. The subset $\mathcal{S}_1$ has $m_0$ samples with $U_t = 0$ and $m_1'$ samples with $U_t = 1$. Denote $\mathcal{D}_0 = \{(u_1, \ldots, u_{t-1}, u_{t+1}, \ldots, u_k) \mid u \in \mathcal{S}_1 \wedge u_t = 0\}$, $\mathcal{D}_1' = \{(u_1, \ldots, u_{t-1}, u_{t+1}, \ldots, u_k) \mid u \in \mathcal{S}_1 \wedge u_t = 1\}$. In orbit $\mathcal{S}_2$, we define $m_1, m_0', \mathcal{D}_1, \mathcal{D}_0'$ similar to in $\mathcal{S}_1$. In Figure B.1 (b), we show the partition $\mathbf{P}^*$ as described in Theorem 1. We will proceed to show $\forall \mathcal{S}_1, \mathcal{S}_2$:

$$d(\mathcal{S}_1) + d(\mathcal{S}_2) \geq d(\mathcal{S}_1^*) + d(\mathcal{S}_2^*). \tag{B.5}$$

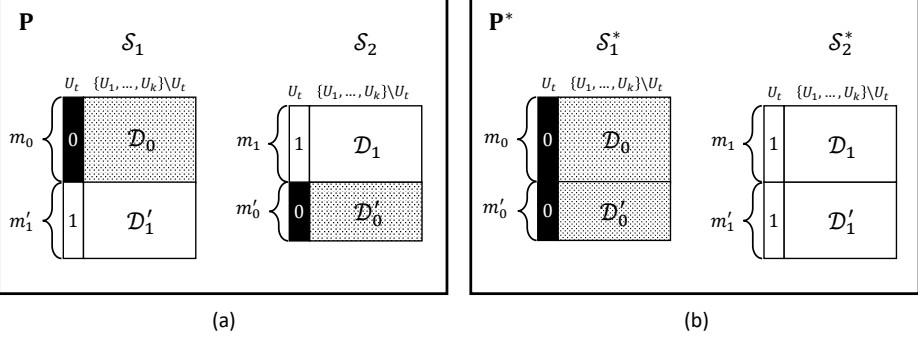

Figure B.1: (a) Any arbitrary data partition $\mathbf{P}$. (b) The partition $\mathbf{P}^*$.

As the hamming distance is calculated per-dimension, we have the following decomposition:

$$d(\mathcal{S}_1) = d(\mathcal{D}_0 \cup \mathcal{D}_1') + \frac{1}{|\mathcal{S}_1|^2} \sum_{u \in \mathcal{S}_1} \sum_{u' \in \mathcal{S}_1} d_H(u_{t+1}, u'_{t+1})$$
$$= d(\mathcal{D}_0 \cup \mathcal{D}_1') + \frac{2m_0 m_1'}{(m_0 + m_1')^2} \tag{B.6}$$

$$d(\mathcal{S}_2) = d(\mathcal{D}_1 \cup \mathcal{D}_0') + \frac{1}{|\mathcal{S}_2|^2} \sum_{u \in \mathcal{S}_2} \sum_{u' \in \mathcal{S}_2} d_H(u_{t+1}, u'_{t+1})$$
$$= d(\mathcal{D}_1 \cup \mathcal{D}_0') + \frac{2m_1 m_0'}{(m_1 + m_0')^2} \tag{B.7}$$

$$d(\mathcal{S}_1^*) = d(\mathcal{D}_0 \cup \mathcal{D}_0') \tag{B.8}$$

$$d(\mathcal{S}_2^*) = d(\mathcal{D}_1 \cup \mathcal{D}_1') \tag{B.9}$$

We will prove Eq. (B.4) by induction. First consider the case where $|\mathcal{D}_1'| = |\mathcal{D}_0'| = 1$. Denote $\mathcal{D}_1' = \{d_1'\}$ and $\mathcal{D}_0' = \{d_0'\}$. We can expand $d(\mathcal{S}_1)$ as

$$d(\mathcal{S}_1) = d(\mathcal{D}_0) + \frac{2}{(m_0 + 1)^2} \sum_{d_0 \in \mathcal{D}_0} d_H(d_0, d_1'). \tag{B.10}$$

We can similarly expand $d(\mathcal{S}_2), d(\mathcal{S}_1^*)$ and $d(\mathcal{S}_2^*)$. Once the same terms are cancelled out, to prove Eq. B.4, we only need to show for any $d_0', d_1'$, we have:

$$m_0 + \sum_{d_0 \in \mathcal{D}_0} d_H(d_0, d_1') + m_1 + \sum_{d_1 \in \mathcal{D}_1} d_H(d_1, d_0') \geq \sum_{d_0 \in \mathcal{D}_0} d_H(d_0, d_0') + \sum_{d_1 \in \mathcal{D}_1} d_H(d_1, d_1'), \tag{B.11}$$

One sufficient condition is that the number of elements in $\mathcal{D}_0, \mathcal{D}_1$ is $2^{t-1}$. This can be empirically achieved with a large dataset. First, we will prove:

$$m_0 + \sum_{d_0 \in \mathcal{D}_0} d_H(d_0, d_1') \geq \sum_{d_0 \in \mathcal{D}_0} d_H(d_0, d_0'). \tag{B.12}$$

Consider $\mathcal{D}_0$ with $2^{t-1}$ unique elements. If the elements in $\mathcal{D}_0$ are such that they lie in a sub-cube of dimension $t - 1$ with $U_j = a$ for $j \in \{1, \ldots, t\}, a \in \{0, 1\}$. Let

$$\Delta = \sum_{d_0 \in \mathcal{D}_0} d_H(d_0, d_0') - \sum_{d_0 \in \mathcal{D}_0} d_H(d_0, d_1') \tag{B.13}$$

It is easy to show that $\Delta$ is maximized when $d_0' \notin \mathcal{D}_0$ and $d_1' \in \mathcal{D}_0$, where $\Delta = m_0$ (example in Figure B.2). This satisfies Eq. (B.11). Now consider a general case of $\mathcal{D}_0$ such that $n_a$ elements have $U_j = a$, forming the set $\mathcal{D}_{0,a}$ and $n_{\bar{a}}$ elements have $U_j = \bar{a}$, forming the set $\mathcal{D}_{0,\bar{a}}$, where $(\bar{\cdot})$ denotes negation. Without loss of generality, let $n_a > n_{\bar{a}}$ (we discuss equality case later). To maximize $\Delta$, $d_0'$ must have $U_j = \bar{a}$, which can be proved through contradiction. If $U_j = a$ for $d_0'$ that maximizes $\Delta$, there exists $d_0''$ which differs from $d_0'$ only on $U_j$, such that $d_H(d_0'', d) - d_H(d_0', d) = 1 \ \forall d \in \mathcal{D}_{0,a}$ and $d_H(d_0'', \bar{d}) - d_H(d_0', \bar{d}) = 1 \ \forall \bar{d} \in \mathcal{D}_{0,\bar{a}}$. As $n_a > n_{\bar{a}}$, we have $\sum_{d_0 \in \mathcal{D}_0} d_H(d_0, d_0') < \sum_{d_0 \in \mathcal{D}_0} d_H(d_0, d_0'')$, which contradicts the condition that we begin with. One can similarly show that $d_1'$ must have $U_j = a$. In the case of equality, both $d_0', d_1'$ can have $U_j = a$ or $U_j = \bar{a}$. Now, starting with the case where every element in $\mathcal{D}_0$ has $U_j = a$ (sub-cube case), for every one additional element in $\mathcal{D}_0$ that we change its $U_j$ to $\bar{a}$ which $n_a >= n_{\bar{a}}$ still holds, $\sum_{d_0 \in \mathcal{D}_0} d_H(d_0, d_0')$ is reduced by 1 and $\sum_{d_0 \in \mathcal{D}_0} d_H(d_0, d_1')$ is increased by 1, which reduces $\Delta$ by 2. Hence $\Delta$ is maximized in the sub-cube case, for which we have already shown Eq. (B.11) holds, and we have proven the sufficient condition.

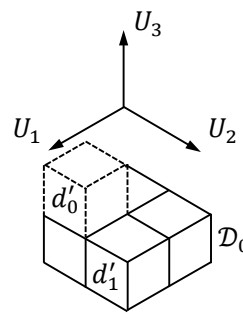

Figure B.2: An example of the sub-cube case with $m_0 = 4$, $t = 3$ and $U_3 = 0$ in $\mathcal{D}_0$. The maximum $\Delta$ is 4 with $U_3 = 1$ for $d_0'$ and $U_3 = 0$ for $d_1'$.

One can apply the same analysis above and prove:

$$m_1 + \sum_{d_1 \in \mathcal{D}_1} d_H(d_1, d_0') \geq \sum_{d_1 \in \mathcal{D}_1} d_H(d_1, d_1'). \tag{B.14}$$

Now we have proved the case where $|\mathcal{D}_0'| = |\mathcal{D}_1'| = 1$. By induction, we assume that Eq. (B.4) holds for $|\mathcal{D}_0'| = |\mathcal{D}_1'| = p, p \geq 1$. We need to prove for the case with one additional element, *i.e.*, $\mathcal{D}_1' \cup d_1''$ and $\mathcal{D}_0' \cup d_0''$. Through reduction, this is to show:

$$m_0 + \sum_{d_0 \in \mathcal{D}_0} d_H(d_0, d_1'') + m_1 + \sum_{d_1 \in \mathcal{D}_1} d_H(d_1, d_0') \geq \sum_{d_0 \in \mathcal{D}_0} d_H(d_0, d_0'') + \sum_{d_1 \in \mathcal{D}_1} d_H(d_1, d_1'). \tag{B.15}$$

This clearly holds given Eq. (B.11) is true.

Eq. (B.4) contains an equality case, we have shown that in the sufficient condition that the equality holds in the sub-cube case, *i.e.*, the partition is such that within every subset, the group action of $c_i, i \in \{1, \ldots, k\}$ on the subspace of $\mathcal{X}$ spanned by the subset is the identity mapping. This means that with the overall SSL loss term itself, the partition may be based on one of $c_1, \ldots, c_k$. However, with the contra-position of the sufficient condition in Theorem 1, the cross-orbit action does not correspond to any $d \in \mathcal{D}$. Hence overall, we have shown the maximization leads to a partition with cross-orbit action as $h \in \mathcal{G}/\mathcal{D}$.

## C  Implementation Details

### C.1  Implementation Details of the CNN Activation Visualization

We used ImageNet100 with image size of 224 for our visualization in Figure 2 (b). The visualization is based on the guided propagation method used in [25], and we adopted the publicly available implementation[1]. We chose VGG-16 [24] as backbone, due to its native support by guided propagation visualization. We followed the default training methods of the SimCLR but only replaced the default ResNet-50 backbone with the VGG16. We trained the baseline and ours model with 200 epochs and for IP-IRM, $\lambda_1 = 0.2$, $\lambda_2 = 0.5$. Please refer to Section C.3.1 for more details. Once the backbone is trained, we first obtained the augmentation-unrelated filters by performing augmentations and removing the filters equivariant to augmentations. Then we performed K-Means clustering ($K = 4$) on the CNN weights of layer 17 and layer 28 among the remaining filters, and chose the 4 filters closest to the cluster center. The motivation for such design is to reveal what the representation captures beyond augmentation-equivariant semantics, and the clustering helps locate the different semantics captured by the representation. We show the activation visualization on layer 18 and 29, *i.e.*, after the ReLU layer, which is the actual input to the next CNN layer, instead of immediately after CNN layer 17 and 28.

---

[1]`https://github.com/utkuozbulak/pytorch-cnn-visualizations`

## C.2 Unsupervised Disentanglement

### C.2.1 Evaluation Metric Details

As we discussed in Section 5.1 of the main paper, here we follow [21, 31] to give more detailed formulas or evaluation methods of the used metrics. All the implementations follow the open-source library [2].

**Disentangle Metric for Informativeness (DCI)**. In [10], the authors propose a complete framework to evaluate disentangled representations instead of a single metric. They report separate scores for modularity, compactness and explicitness, which they call disentanglement, completeness and informativeness. It computes the importance of each dimension of the learned representation for predicting a factor of variation. The predictive importance $R_{ij}$ of the dimensions of feature can be computed with a Lasso or a Random Forest classifier. For the lasso regressor, the importance weights $R_{ij}$ are the magnitudes of the weights learned by the model, while the Gini importance [2] of code dimensions is used with random forests. Here we adopted the Informativeness score, which can be computed as the prediction error of predicting the factors of variations.

**Interventional Robustness Score (IRS)**. IRS [26] propose to measure the feature robustness by computing distances between sets of codes before and after an intervention on factor realizations. The intuition behind the metric is that changes in nuisance factors should not impact code dimensions attributed to targeted factors. First a reference set is created from instances where realizations of target factors are fixed. Then a second set contains instances with the same targeted factor realization, but different realizations of nuisance factors. The metric computes the distance between the mean of code dimensions associated to targeted factors. This sampling and distance measurement procedure is repeated several times and the maximum observed distance is reported to reflect the worst case. The final metric reports a weighted average of the maximum distances. The distances are weighted by the frequency of the factor realizations in the data set.

**Modularity Score (MOD) & Explicitness Score (EXP)**. In [23], for modularity, the authors estimate the mutual information between each code dimension and each factor. If a code dimension is ideally modular, it will have high mutual information with a single factor and zero mutual information with all other factors. Given a single code dimension $i$ and a factor $f$, we denote the mutual information between the code and factor by $m_{if}$. For ease to computation, authors also create a vector $\mathbf{t}_i$ fo the same size, which represetns the best-matching case of ideal modularity for code dimension $i$:

$$t_{if} = \begin{cases} \theta_i, & \text{if } f = \arg\max_g (m_{ig}) \\ 0, & \text{otherwise,} \end{cases}$$

where $\theta_i = \max_g (m_{ig})$. The observed deviation from the template is given by

$$\delta_i = \frac{\sum_f (m_{if} - t_{if})^2}{\theta_i^2 (N-1)},$$

where $N$ is the number of factors. A deviation of 0 indicates that we have achieved perfect modularity and 1 indicates that this dimension has equal mutual information with every factor. Thus, finally $1 - \delta_i$ is used as a modularity score for code dimension $i$ and the mean of $1 - \delta_i$ over $i$ as the modularity score for the overall code. For explicitness, the authors propose to use a classifier trained on the entire latent codes to predict factor classes, assuming that factors have discrete values. They suggest using a simple classifier such as logistic regression and report classification performance using the ROC area-under-the-curve (AUC). The final score is the average AUROC over all classes for all factors.

**Downstream Tasks with LR and GBT**. We follow [21] to consider the simplest downstream classification task where the goal is to recover the true factors of variations from the learned feature using either multi-class logistic regression (LR) or gradient boosted trees (GBT). For each factor we fit a different model and then report the average test accuracy across factors. We consider two different models. First, we train a cross validated logistic regression from Scikit-learn with 10 different values for the regularization strength ($Cs = 10$) and 5 folds. Finally, we train a gradient boosting classifier from Scikit-learn with default parameters. We sample the training set of 7500 and the evaluation set of 2500 samples for CMNIST; while the training set of 1000 and the evaluation set of 2500 for Shapes3D.

---

[2]`https://github.com/google-research/disentanglement_lib`

### C.2.2 Model Architecture

In the main paper we stated that we used CNN-based feature extractor bascknones with comparable number of parameters for all the baselines and IP-IRM. Here we provide the detailed model architectures in Table C.1, C.2, C.3, C.4.

| Encoder | Decoder |
|---|---|
| Input: $28 \times 28 \times 3$ | Input: $\mathbb{R}^{10}$ |
| FC, $2352 \times 148$ ReLU | FC, $10 \times 148$ ReLU |
| FC, $148 \times 148$ ReLU | FC, $148 \times 148$ ReLU |
| FC, $148 \times 20$ | FC, $148 \times 2352$ Sigmoid |

Table C.1: Encoder and Decoder architecture of the VAE-based methods with 0.747M parameters for CMNIST dataset in the main experiment.

| Encoder | Decoder |
|---|---|
| Input: $64 \times 64 \times 3$ | Input: $\mathbb{R}^{10}$ |
| $4 \times 4$ conv, 32 ReLU, stride 2, padding 1 | FC, $10 \times 32$ ReLU |
| $4 \times 4$ conv, 32 ReLU, stride 2, padding 1 | FC, $32 \times 512$ ReLU |
| $4 \times 4$ conv, 32 ReLU, stride 2, padding 1 | $4 \times 4$ upconv, 32 ReLU, stride 2, padding 1 |
| $4 \times 4$ conv, 32 ReLU, stride 2, padding 1 | $4 \times 4$ upconv, 32 ReLU, stride 2, padding 1 |
| FC, $512 \times 32$ ReLU | $4 \times 4$ upconv, 32 ReLU, stride 2, padding 1 |
| FC, $32 \times 20$ | $4 \times 4$ upconv, 3 Sigmoid, stride 2, padding 1 |

Table C.2: Encoder and Decoder architecture of the VAE-based methods with 0.136M parameters for Shapes3D dataset in the main experiment.

| Encoder |
|---|
| Input: $28 \times 28 \times 3$ |
| FC, $2352 \times 256$ ReLU |
| FC, $256 \times 256$ ReLU |
| FC, $256 \times 10$ |

| Projection Head |
|---|
| FC, $10 \times 64$ BatchNorm ReLU |
| FC, $64 \times 32$ |

Table C.3: Model architecture of the SSL models with 0.674M parameters for CMNIST dataset in the main experiment.

### C.2.3 Training Details

For all the methods, the training epoch is fixed to 200 and training batch size is fixed to 2048. The optimizer is Adam and feature dimension was set to 10. Note that due to much more computational complexity of the self-supervised learning, we *decreased* the sample size used for training and evaluation with both VAE and SSL models to make sure a fair comparison. Specifically, for CMNIST, we used 50,000 samples for training and 10,000 for testing. For Shapes3D, we used 90,000 for training and 10,000 for testing. For VAE-based model, we followed their common hyperparameters setting without fine-tuning and the implementation codes were built upon the open-source code [3]. Specifically, the learning rate is 5e-4 for all the models except for the 1e-4 of Factor-VAE. For $\beta$-VAE, $\beta = 4$. For $\beta$-AnnealVAE, the starting annealed capacity $C = 0$ and the final annealed capacity $C = 25$. For $\beta$-TCVAE, $\beta = 6, \alpha = 1, \gamma = 1$. For Factor-VAE, $\gamma = 6$, the learning rate of the

---

[3] `https://github.com/YannDubs/disentangling-vae`

| Encoder |
| --- |
| Input: $64 \times 64 \times 3$ |
| $4 \times 4$ conv, 32 BatchNorm ReLU, stride 2, padding 1 |
| $4 \times 4$ conv, 32 BatchNorm ReLU, stride 2, padding 1 |
| $4 \times 4$ conv, 64 BatchNorm ReLU, stride 2, padding 1 |
| $4 \times 4$ conv, 64 BatchNorm ReLU, stride 2, padding 1 |
| $4 \times 4$ Average Pooling |
| FC, $64 \times 10$ |

| Projection Head |
| --- |
| FC, $10 \times 64$ BatchNorm ReLU |
| FC, $64 \times 32$ |

Table C.4: Model architecture of the SSL models with 0.120M parameters for Shapes3D dataset in the main experiment.

discriminator is 5e-5 with Adam optimizer. For SSL-based methods, we used SimCLR as our base model. The learning rate is set to 1e-3 and temperature is 0.5. For IP-IRM, $\lambda_1 = 0.2, \lambda_2 = 0.5$, the partition $\mathbf{P}$ is updated every 30 epochs and optimized from random every time. All the experiments were completed on the work station with 4 Nvidia 2080Ti GPUs.

## C.3 Self-supervised Learning

### C.3.1 Implementation Details

For SimCLR, we follow [8, 16] to use ResNet-50 as the encoder architecture and use the Adam optimizer. The temperature is set to 0.5 and the dimension of the latent vector is 128. All the models were trained for 400 or 1000 epochs and evaluated by training a linear classifier after fixing the learned features. The learning rate is set to 0.001 and weight decay is 1e-6 for both SSL pretraining and downstream fine-tuning. For DCL [8] and HCL [16], we adopted the best parameters posted in the original paper and followed the implementation from the open-source code [4]. Particularly, for DCL, $\tau^+ = 0.12$ on STL10 and Cifar100 dataset; while for HCL, $\tau^+ = 0.1, \beta = 1$ on STL10, $\tau^+ = 0.05, \beta = 0.5$ on Cifar100. For our IP-IRM, we applied $\lambda_1 = 0.2, \lambda_2 = 0.5$ on STL10 and $\lambda_1 = 0.2, \lambda_2 = 0.2$ on Cifar100. We additionaly performed baselines and our IP-IRM on ImageNet-100 [27], a randomly chosen subset of 100 classes of ImageNet with 200 training epochs. The results are reported in Table C.6. We followed the best parameters reported in the paper [8, 16]. Particularly, for DCL, $\tau^+ = 0.01$. For HCL, $\tau^+ = 0.01, \beta = 1.0$. Note that on ImageNet100, we slightly modified the $\tau$ and $\beta$ to achieve the better performance. Specifically, for DCL+IP-IRM, we set $\tau = 0.1$; while for HCL+IP-IRM, we set $\tau = 0.1, \beta = 0.5$. While extending ssl pretraining process to 1000 epochs with MixUp [19], we directly followed the open-source MixUp implementation[5] for SimCLR: the temperature is set to 0.2 and MixUp alpha is set to 1.0. For our IP-IRM, note that the MixUp was processed within each subsets to avoid the sample confusion of two subsets. Therefore, the training time inevitably grow linearly due to the increase of the partitions. To mitigate this problem, we controlled the number of partitions (*e.g.*, 5 partitions, FIFO) in practice as an approximation. For the supervised training, as introduced in the main paper, we adopted the same codebase, optimizer and parameter setting, *i.e.*, the Adam optimizer with learning rate as 0.001 and weight decay as 1e-6 for 100 epochs. We only added the learning rate decay to achieve the optimal training at 60 and 80 epoch. Moreover, we found that adding MixUp with $\alpha = 1.0$ would decrease the performance. Therefore, we set $\alpha$ as 0.5. The experiments were completed on the workstation with 4 Nvidia 2080Ti GPUs.

When training on the large-scale ImageNet, we built our IP-IRM based on SimCLR, MoCo-v2 and SimSiam [7] [6]. The batch size is 512 for SimCLR due to the limited computation resources. Different

---

[4] https://github.com/chingyaoc/DCL, https://github.com/joshr17/HCL

[5] https://github.com/kibok90/imix

[6] https://github.com/taoyang1122/pytorch-SimSiam (Note that the official implementation of SimSiam was not available then.)

| Class \ Context | Long-Tailed Contexts | | | | | | | Zero-Shot Contexts | | |
|---|---|---|---|---|---|---|---|---|---|---|
| Dog | on grass | in water | in cage | eating | on beach | lying | running | at home | in street | on snow |
| Cat | on snow | at home | in street | walking | in river | in cage | eating | in water | on grass | on tree |
| Bear | in forest | black | brown | eating grass | in water | lying | on snow | on ground | on tree | white |
| Sheep | eating | on road | walking | on snow | on grass | lying | in forest | aside people | in water | at sunset |
| Bird | on ground | in hand | on branch | flying | eating | on grass | standing | in water | in cage | on shoulder |
| Rat | at home | in hole | in cage | in forest | in water | on grass | eating | lying | on snow | running |
| Horse | on beach | aside people | running | lying | on grass | on snow | in forest | at home | in river | in street |
| Elephant | in zoo | in circus | in forest | in river | eating | standing | on grass | in street | lying | on snow |
| Cow | in river | lying | standing | eating | in forest | on grass | on snow | at home | aside people | spotted |
| Monkey | sitting | walking | in water | on snow | in forest | eating | on grass | in cage | on beach | climbing |

Table C.5: Construction of our NICO [14] subset for OOD multi-classification . **Context** denotes the context class name, while **Class** represents the object class name. "Long-Tailed Contexts" is the training contexts arranged by the sample number order (from more to less) and "Zero-shot Contexts" represents the context labels only appear in testing rather than training.

with the contrastive loss adopted in SimCLR and other conventional SSL methods, SimSiam discards the negative samples and uses the MSE loss. Therefore, our IP-IRM was built directly based on MSE loss and encourages the samples in each subset achieve the same performance (*i.e.*, the same MSE loss). For the detailed proof that IRM can be applied to any convex loss function, please refer to [1]. $\lambda_1 = 0.2, \lambda_2 = 0.5$, partition $\mathbf{P}$ was updated every 50 epochs. Other hyper-parameters followed the default SimSiam setting. For downstream linear classifier training, we followed the open-source code to use Nvidia LARC optimizer with learning rate 1.6. The experiments were completed on the workstation with 8 Nvidia V100 GPUs.

## C.4 OOD Classification on NICO

### C.4.1 NICO dataset

In our experiment, we selected a subset of NICO animal dataset [14] as a challenging benchmark to test the feature decomposability for proposed IP-IRM and baselines. Specifically, images in NICO are labeled with a context background class (*e.g.*, "on grass"), besides the object foreground class (*e.g.*, "dog"). For each animal class, we randomly sample its images and make sure the context labels of those images are within a fixed set of 10 classes (e.g.,"snow", "on grass" and "in water"). Based on these data, we propose a challenging OOD setting including three factors regarding contexts: 1) Long-Tailed: training context labels are in long-tailed distribution in each individual class, e.g., "sheep" might have 10 images of "on grass", 5 images of "in water" and 1 image of "on road"; 2) Zero-Shot: for each object class, 7 out of 10 context labels are in training images and the other 3 labels appear only in testing; 3) Orthogonal—the head context label of each object class is set to be as unique (dominating only in one object class) as possible.

The detailed 7 context classes (long-tailed contexts) and 3 zero-shot contexts are shown in Table C.5) for each object class. Next, we formed a long-tailed training dataset by selecting part of the images in each context class with multiplying a ratio. In particular, the ratio for $w$-th context class ($w \in \{0, \ldots, 6\}$) is given by

$$\text{ratio} = \text{IR}^{w/6}, \quad \text{(C.1)}$$

where $\text{IR}$ is a hyper-parameter that denotes the imbalance ratio. The effect of $\text{IR}$ on ratio is shown in Figure C.1 — lower ratio leads to the harder OOD problem. In the main paper we keep $\text{IR} = 0.02$. During testing, the number of test samples across the 7 context classes are balanced, *i.e.*, 50 samples per context. Moreover, we added 3 zero-shot context classes for each

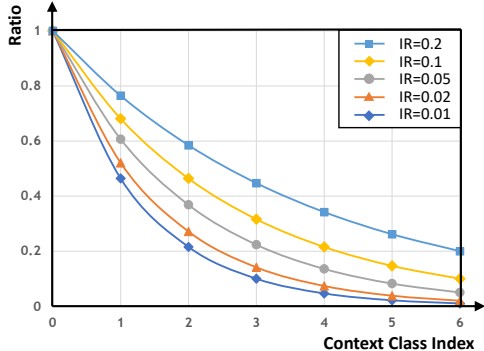

Figure C.1: Plot of context class index against its corresponding ratio under various imbalance ratio (IR).

object class as shown in Table C.5 (last three columns). These zero-shot context classes have the larger number of test samples (100 samples per context). Therefore, a model that performs well in

our split must be robust to both long-tailed and zero-shot problems *w.r.t.* the context class. Figure C.2 shows an example of our constructed subset for "cat" and "dog" during training and testing.

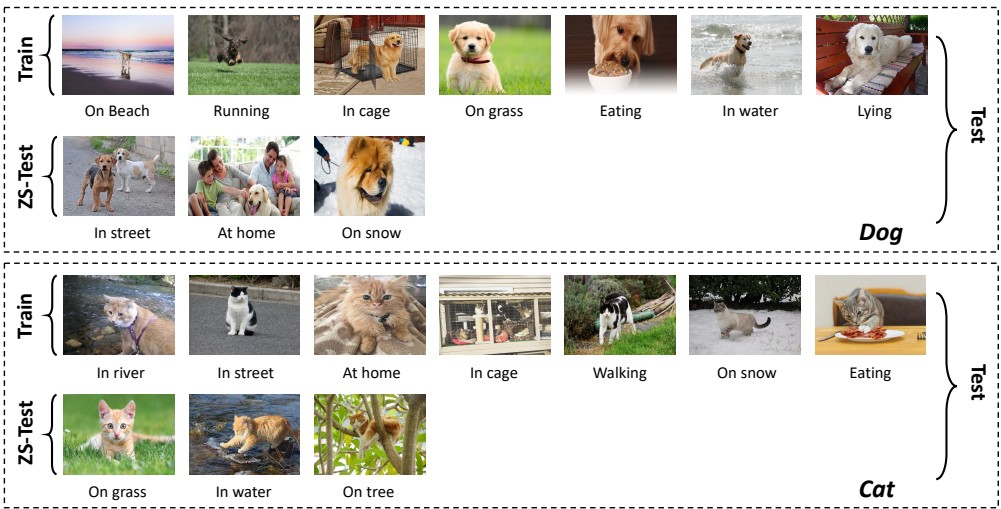

Figure C.2: We list the sample images of each context class using "Dog" and "Cat" as the example in our constructed NICO dataset. **Train**, **Test** and **ZS-Test** denote samples for training, testing and zero shot testing respectively. Note that there is no overlap between training and testing images.

### C.4.2 Training Details

Different from the linear classifier fine-tuning of SSL which uses a linear fc layer mapping from the feature dimension to the class label dimension, we adopted a linear fc layer mapping from the feature dimension to the feature dimension (*e.g.*, $\mathbb{R}^{2048 \times 2048}$) as the feature space mapping for bias NICO training. Then the classification (both training and inference) is based on the metric learning paradigm (*e.g.*, supervised contrastive learning) by measuring the distance between sample features (*e.g.*, $k$-nn accuracy). The reason is that the shared feature mapping layer can help classifier to neglect the bias feature. Empirically, this paradigm outperforms the conventional classifier by more than 10%. For parameter setting details, the learning rate is set to 0.2 with SGD optimizer and the batch size is fixed to 128. We utilized $k$-nn classifier ($k = 10$) for evaluation. All the models are trained for 150 epochs and the first 2 epochs are the warm-up stage. Learning rate was decreased by 5 at 80, 120 epoch. We report the best accuracy during training as the final performance.

### C.5 Transfer Learning

We used the ResNet-50 backbone trained on ImageNet through SSL or our IP-IRM as the feature extractor when transferring to the downstream tasks. For the baseline models, we followed [11] to download the pre-trained weights of the ResNet50 models in the open-source code. All models have 23.5M parameters in their backbones and were pre-trained on the ImageNet [9] training set, consisting of 1.28M images.

### C.5.1 Many-shot Learning

The top-1 accuracy metric is reported on Food-101, CIFAR-10, CIFAR-100, SUN397, Stanford Cars, and DTD, mean per-class accuracy on FGVC Aircraft, Oxford-IIIT Pets, Caltech-101, and Oxford 102 Flowers and the 11-point mAP metric on Pascal VOC 2007. On Caltech-101 we randomly selected 30 images per class to form the training set and we test on the rest. We used the first train/test split defined in DTD and SUN397. On FGVC Aircraft, Pascal VOC2007, DTD, and Oxford 102 Flowers we used the validation sets defined by the authors, and on the other datasets we randomly select 20% of the training set to form the validation set. The optimal hyperparameters were selected on the validation set, after which we retrained the model on all training and validation images. Finally, the accuracy is computed on the test set. For the downstream transfer learning, we finetuned the models following [11] with minor modifications. We train for 5000 steps with

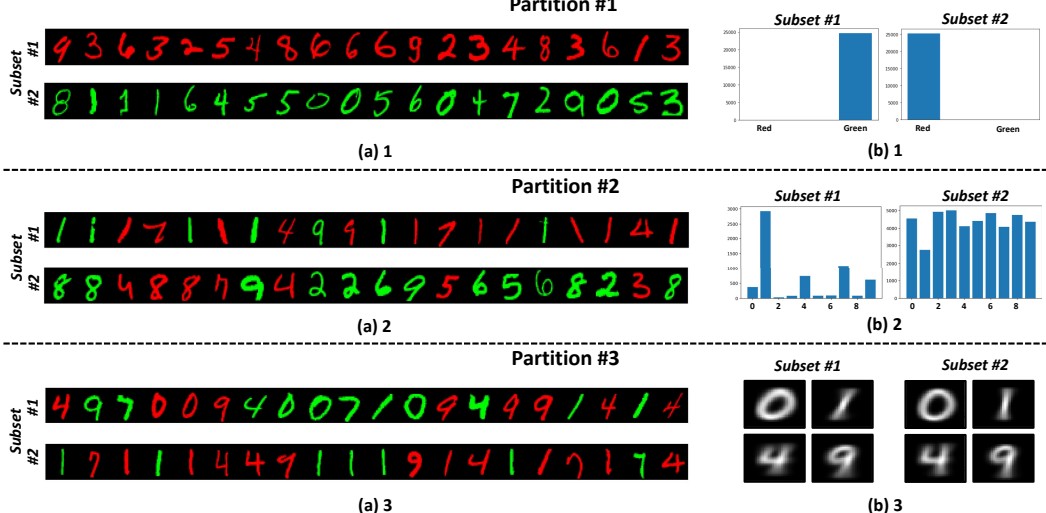

Figure C.3: Visualization of the obtained partition $\mathbf{P}^*$ during training on the full CMNIST dataset with balanced colorization. The left part (*i.e.*, (a)1, (a)2, (a)3) is the 20 samples chosen randomly from each specific subset in different partitions, similar to the Figure 5 in the main paper. Note that for (a)3 we mainly show 0,1,4,7,9 for ease of comparing. The right part ((b)1, (b)2, (b)3) is the global statistic view of the two subsets. Specifically, (b)1 plots the number of images with different colors in subset #1 and #2; (b)2 draws the number of images in terms of the digit in subset #1 and #2; (b)3 shows the images by averaging samples from the same digit.

a batch size of 64. The optimiser is SGD with Nesterov momentum and a momentum parameter of 0.9. The learning rate follows a cosine annealing schedule without restarts, and the initial learning rate is chosen from a grid of 4 logarithmically spaced values between 0.0001 and 0.1. The weight decay is similarly chosen from a grid of 4 logarithmically spaced values between 1e-6 and 1e-3, along with no weight decay. These weight decay values are divided by the learning rate. We selected the data augmentation from: random crop with resize and flip, or simply a center crop.

### C.5.2 Few-shot Learning

For the dataloader, no augmentation is used and the images are resized to 224 pixels along the shorter side using bicubic resampling, followed by

| Method | ImageNet100 | | | |
| | *k*-NN | | Linear | |
| | *Top-1* | *Top-5* | *Top-1* | *Top-5* |
|---|---|---|---|---|
| SimCLR [5] | 64.64 | 89.36 | 76.58 | 94.22 |
| DCL [8] | 65.12 | 89.86 | 77.32 | 94.74 |
| HCL [16] | 69.24 | 90.66 | 79.15 | 94.72 |
| **SimCLR+IP-IRM** | 69.08 | 90.96 | 79.52 | 94.76 |
| **DCL+IP-IRM** | 69.62 | 91.62 | **80.38** | 95.32 |
| **HCL+IP-IRM** | **71.78** | **91.94** | 80.30 | **95.38** |

Table C.6: Accuracy (%) of k-NN and linear classifiers trained on the representations learnt on ImageNet100 [27] with 200 training epochs. We used the pretext task in SimCLR [5], DCL [8] and HCL [16] with IP-IRM, denoted as $(\cdot)$ + IP-IRM.

a center crop of $224 \times 224$. For all methods on all datasets, we trained a linear classifier in each episode using SGD optimizer with learning rate as 0.01 and weight decay as 0.001. The classifier was trained for 100 epochs with batch size as 4. The learned classifier is evaluated using 15 query images in each episode and the reported accuracies and errors are computed on 2000 total episodes.

## D  Additional Experimental Results

### D.1  Visualization of Partition $\mathbf{P}^*$

As we proposed in Section 5.1, here we plot the partition results on the full CMNIST dataset with balanced colorization (each image is uniformly colored by red or green) in Figure C.3. We can see in each partition, the obtained $\mathbf{P}^*$ still tells apart a specific semantic into two subsets. Similar to that in the experiment on CMNIST binary dataset (see Figure 5 (a) in the main paper), the semantics of color, digit and slant can be obviously discovered by our IP-IRM algorithm.

| | Method | DCI | IRS | MOD | EXP | LR | GBT | Average |
|---|---|---|---|---|---|---|---|---|
| **Shapes3D (Full)** | VAE [18] | 0.473±0.021 | 0.525±0.009 | **0.907**±0.009 | 0.895±0.025 | **0.753**±0.072 | 0.472±0.021 | 0.671±0.015 |
| | β-VAE [15] | 0.495±0.023 | 0.485±0.032 | 0.859±0.026 | **0.899**±0.006 | 0.713±0.013 | 0.496±0.025 | 0.658±0.012 |
| | β-AnnealVAE [3] | **0.634**±0.033 | **0.813**±0.050 | 0.758±0.095 | 0.786±0.024 | 0.598±0.034 | **0.634**±0.033 | **0.704**±0.025 |
| | β-TCVAE [4] | 0.556±0.048 | 0.524±0.074 | 0.797±0.054 | 0.897±0.028 | 0.718±0.042 | 0.569±0.072 | 0.677±0.048 |
| | Factor-VAE [17] | 0.454±0.013 | 0.447±0.027 | 0.798±0.032 | 0.839±0.030 | 0.637±0.029 | 0.455±0.013 | 0.605±0.007 |
| | SimCLR [5] | 0.368±0.020 | 0.452±0.031 | 0.795±0.034 | 0.796±0.008 | 0.516±0.055 | 0.369±0.019 | 0.549±0.027 |
| | **IP-IRM (Ours)** | 0.392±0.010 | 0.426±0.011 | 0.835±0.024 | 0.806±0.006 | 0.523±0.018 | 0.391±0.012 | 0.562±0.012 |

Table D.1: Results on disentanglement metrics of existing unsupervised disentanglement methods, standard SSL (SimCLR [5]) and IP-IRM using Shapes3D [17] (full). Results are averaged over 4 trails (mean ± std).

## D.2 Unsupervised Disentanglement on Full Shapes3D Dataset

As we introduced in Section 5.1, we evaluate our IP-IRM and baseline models on Shapes3D dataset with only first three semantics, as the standard augmentations in SSL will contaminate any color-related semantics. Here we present the results on full Shapes3D dataset in Table D.1. We can find that the VAE-based models indeed perform much better than the SSL model due to the well-dientangled color semantics.

## D.3 Additional Results on ImageNet100

We additionally performed our algorithm and baselines on the medium-scale dataset — ImageNet100. Similar to the results presented in the Table 2 of the main paper, incorporating IP-IRM algorithm to the baselines still brings the huge performance boosts to both $k$-NN and linear classification. For example, we can observe that our IP-IRM improves the SimCLR by 4.44% on $k$-NN Top-1 and 2.94% on linear classification Top-1 accuracy. Moreover, our DCL+IP-IRM achieves a new state-of-the-art accuracy of 80.38% in the downstream linear classification evaluation. Also we can find that IP-IRM brings more performance gain with $k$-NN classifier compared to the supervised linear classifier, *e.g.*, 4.50% using DCL+IP-IRM and 2.54% with HCL+IP-IRM. This validates that the constructed feature space using IP-IRM more faithfully reflect the semantic differences than the baselines.

## D.4 Detailed Results of Transfer Learning

| | Method | Aircraft | Caltech | Cars | Cifar10 | Cifar100 | DTD | Flowers | Food | Pets | SUN | VOC | Avg. |
|---|---|---|---|---|---|---|---|---|---|---|---|---|---|
| **Many-Shot** | InsDis [29] | 36.87 | 71.12 | 28.98 | 80.28 | 59.97 | 68.46 | 83.44 | 63.39 | 68.78 | 49.47 | 74.37 | 62.29 |
| | PCL [20] | 21.61 | 76.90 | 12.93 | 81.84 | 55.74 | 62.87 | 64.73 | 48.02 | 75.34 | 45.70 | 78.31 | 56.73 |
| | PIRL [22] | 37.08 | 74.48 | 28.72 | 82.53 | 61.26 | 68.99 | 83.60 | 64.65 | 71.36 | 53.89 | 76.61 | 63.92 |
| | MoCo-v1 [13] | 35.55 | 75.33 | 27.99 | 80.16 | 57.71 | 68.83 | 82.10 | 62.10 | 69.84 | 51.02 | 75.93 | 62.41 |
| | MoCo-v2 [6] | 41.28 | **87.91** | 40.04 | 91.33 | 73.14 | **74.47** | 89.02 | 67.10 | 80.49 | 58.10 | 80.13 | 71.18 |
| | **IP-IRM (Ours)** | **43.12** | 87.22 | **41.16** | **91.84** | **74.13** | 73.94 | **89.23** | **68.05** | **81.70** | **58.41** | **80.32** | **71.74** |
| **20-Shot** | InsDis [29] | 47.44 | 88.32 | 54.37 | 67.60 | 72.79 | 82.37 | 92.98 | 69.49 | 82.84 | 90.08 | - | 74.82 |
| | PCL [20] | 44.72 | 92.42 | 47.55 | 69.13 | 70.95 | 79.55 | 81.82 | 67.57 | 92.30 | 90.25 | - | 73.63 |
| | PIRL [22] | **48.15** | 90.01 | 55.20 | 67.87 | 73.27 | 82.83 | 92.89 | 69.98 | 84.43 | 90.54 | - | 75.52 |
| | MoCo-v1 [13] | 47.08 | 90.81 | 51.98 | 64.01 | 69.72 | 83.26 | 92.54 | 70.12 | 83.78 | 90.00 | - | 74.33 |
| | MoCo-v2 [6] | 42.57 | 95.75 | 57.30 | 73.99 | 79.20 | **86.83** | 93.15 | 73.64 | 91.59 | 93.85 | - | 78.78 |
| | **IP-IRM (Ours)** | 43.92 | **95.77** | **57.59** | **76.43** | **81.56** | 86.69 | **94.03** | **74.81** | **92.35** | **94.19** | - | **79.73** |

Table D.2: Accuracy (%) of transfer learning experiments using representation trained on ImageNet [9]. Few-shot experiment was conducted with 5-way-20-shot setting using 2,000 episodes. We excluded VOC [12] in few-shot experiments following [11] as it is a multi-label dataset where standard few-shot evaluation is not applicable.

As we wrote in the main paper that we also conducted experiments of many-shot and 20-shot, here we show the results in Table D.2. From the results we can make the following observations: (i) The downstream transfer performance is approximately correlated to the in-domain self-supervised learning evaluation, *i.e.*, the linear classification accuracy. The better methods in ImageNet classification can also obtain higher transfer performance. (ii) Our IP-IRM outperforms all the counterparts on most datasets and achieves best performance on the Avg.. (iii) Combining with the 5-shot results in Table 4 of the main paper, we can find with the increasing training samples (*i.e.*, 5-shot → 20-shot → many-shot), the performance gain of our IP-IRM decreases. This is in line with the conclusion of [28] that the disentangled feature is more helpful for the learning with fewer training samples.

| Method | Aircraft | Caltech | Cars | Cifar10 | Cifar100 |
|---|---|---|---|---|---|
| InsDis [29] | 35.07±0.43 | 75.97±0.47 | 37.49±0.36 | 51.49±0.40 | 57.61±0.48 |
| PCL [20] | **36.86**±0.44 | 90.72±0.30 | 39.68±0.39 | 59.26±0.36 | 60.78±0.46 |
| PIRL [22] | 36.70±0.44 | 78.63±0.46 | 39.21±0.37 | 49.85±0.42 | 55.23±0.52 |
| MoCo-v1 [13] | 35.31±0.44 | 79.60±0.45 | 36.35±0.35 | 46.96±0.39 | 51.62±0.50 |
| MoCo-v2 [6] | 31.98±0.38 | 92.32±0.29 | 41.47±0.41 | 56.50±0.42 | 63.33 ±0.51 |
| **IP-IRM (Ours)** | 32.98±0.40 | **93.16**±0.26 | **42.87**±0.41 | **60.73**±0.39 | **68.54**±0.49 |

| Method | DTD | Flowers | Food | Pets | SUN |
|---|---|---|---|---|---|
| InsDis [29] | 69.38±0.43 | 77.35±0.51 | 50.01±0.45 | 66.38±0.44 | 74.97±0.48 |
| PCL [20] | 69.53±0.43 | 67.50±0.50 | 57.06±0.44 | **88.31**±0.36 | 84.51±0.37 |
| PIRL [22] | 70.43±0.42 | 78.37±0.48 | 51.61±0.45 | 69.40±0.43 | 76.64±0.46 |
| MoCo-v1 [13] | 68.76±0.46 | 75.42±0.53 | 49.77±0.46 | 68.32±0.43 | 74.77±0.50 |
| MoCo-v2 [6] | 78.00±0.38 | 80.05±0.45 | 57.25±0.48 | 83.23±0.40 | 88.10±0.33 |
| **IP-IRM (Ours)** | **79.30**±0.36 | **82.68**±0.41 | **59.61**±0.46 | 85.23±0.38 | **89.38**±0.30 |

Table D.3: Accuracy (%) of 5-way-5-shot few-shot evaluation with standard deviation (mean±std) using the image representation learned on ImageNet [9].

To report the error bars as stated in the checklist, here we presented the standard deviation for the 5-shot experiments as a supplement for the Table 4 in the main paper. As shown in Table D.3, all the methods perform in a stable interval.

### D.5 Interventional Few-Shot Learning

We evaluated the learned representations by SSL and our IP-IRM with the IFSL classifier proposed in [30], which is an intervention-based classifier designed to remove the confounding bias from the misuse of pre-trained knowledge in few-shot learning (*e.g.*, treating grass as dog when most dogs appear with grass in the few-shot training images). We adopted the combined adjustment implementation with $n = 4$ (dividing feature channels into 4 subsets). For all models in all datasets, we trained the IFSL classifier using the Adam optimizer with learning rate as 0.01 for 50 epochs. The batch size was set as 4. The reported accuracy is based on the average across 2,000 episodes.

| | Method | Aircraft | Caltech | Cars | Cifar10 | Cifar100 | DTD | Flowers | Food | Pets | SUN | Avg. |
|---|---|---|---|---|---|---|---|---|---|---|---|---|
| Linear | InsDis [29] | 35.07 | 75.97 | 37.49 | 51.49 | 57.61 | 69.38 | 77.35 | 50.01 | 66.38 | 74.97 | 59.57 |
| | PCL [20] | **36.86** | 90.72 | 39.68 | 59.26 | 60.78 | 69.53 | 67.50 | 57.06 | **88.31** | 84.51 | 65.42 |
| | PIRL [22] | 36.70 | 78.63 | 39.21 | 49.85 | 55.23 | 70.43 | 78.37 | 51.61 | 69.40 | 76.64 | 60.61 |
| | MoCo-v1 [13] | 35.31 | 79.60 | 36.35 | 46.96 | 51.62 | 68.76 | 75.42 | 49.77 | 68.32 | 74.77 | 58.69 |
| | MoCo-v2 [6] | 31.98 | 92.32 | 41.47 | 56.50 | 63.33 | 78.00 | 80.05 | 57.25 | 83.23 | 88.10 | 67.22 |
| | **IP-IRM (Ours)** | 32.98 | **93.16** | **42.87** | **60.73** | **68.54** | **79.30** | **82.68** | **59.61** | 85.23 | **89.38** | **69.44** |
| IFSL | InsDis [29] | 41.49 | 86.53 | 45.37 | 60.51 | 66.94 | 75.99 | 87.61 | 58.85 | 80.12 | 85.87 | 68.93 |
| | PCL [20] | 41.22 | 91.23 | 44.55 | 64.96 | 68.10 | 73.10 | 77.99 | 62.35 | 90.88 | 87.28 | 70.17 |
| | PIRL [22] | **42.31** | 88.54 | 46.61 | 61.07 | 66.49 | 76.70 | **88.20** | 60.37 | 82.59 | 87.01 | 69.99 |
| | MoCo-v1 [13] | 41.43 | 89.30 | 44.20 | 57.01 | 63.29 | 76.84 | 87.12 | 60.80 | 81.73 | 85.90 | 68.76 |
| | MoCo-v2 [6] | 40.02 | 93.95 | 51.12 | 69.56 | 73.84 | 80.67 | 86.94 | 66.10 | 90.38 | 91.18 | 74.38 |
| | **IP-IRM (Ours)** | 41.49 | **94.31** | **52.80** | **72.02** | **76.52** | **80.78** | 87.69 | **67.35** | **91.13** | **91.62** | **75.57** |

Table D.4: Accuracy (%) of IFSL classifier using representation trained on ImageNet [9]. Experiment was conducted with 5-way-5-shot setting using 2,000 episodes. We used the combined adjustment for all experiments.

From the results shown in Table D.4, we can observe that: i) our IP-IRM obtains the best accuracy, validating the superiority of the learned disentangle representation. ii) IFSL significantly improves the performance of the vanilla linear classifier, showing that IFSL can further deconfound. iii) Compared with linear classifier, the performance gain of IP-IRM over the best baseline (MoCo-v2) is not as high (1.19% v.s. 2.22%). This is reasonable, since the disentangled feature with our proposed IP-IRM helps the downstream classifier to deconfound by neglecting the non-discriminative features (see Section 5.3).

## E Others

### E.1 License

We use the following open-source database and the license just follow them.

- `https://github.com/YannDubs/disentangling-vae`
- `https://github.com/chingyaoc/DCL`
- `https://github.com/joshr17/HCL`
- `https://github.com/facebookresearch/moco`
- `https://github.com/taoyang1122/pytorch-SimSiam`
- `https://github.com/facebookresearch/InvariantRiskMinimization`
- `https://github.com/kibok90/imix`

### E.2 Data Consent

We entirely use the open-source data and they have already obtained the consent during the data collection. Details can be checked on their website.

### E.3 Personally Identifiable Information

The open-source data we used does not contain personally identifiable information or offensive context. More information can refer to the website of the open-source data.