# OpenReview forum: "Self-Supervised Learning Disentangled Group Representation as Feature"
_NeurIPS.cc/2021/Conference — NeurIPS 2021 Spotlight_

### Official Review · Reviewer_2DAu · 2021-07-14

**Rating:** 8
**Confidence:** 4

**Summary:**

This paper proposes a self-supervised learning model which is a variant of the Invariant Risk Minimization with an iterative partition mechanism. The method can encourage the self-supervised learning of a disentangled representation. Empirical results validate the effectiveness of the proposed model on self-supervised learning and disentangled representation learning.

**Limitations And Societal Impact:**

Yes.

**Main Review:**

Originality:
The proposed method depends on the Invariant Risk Minimization model. The disentanglement validation depends on the symmetry-based (SB) disentangled representation definition. The overall learning setting is the same as the general self-supervised learning. The core technical originality I personally believe is the iterative partition mechanism, but I would also consider the combination of IRM, SSL, and SB-disentanglement is novel and original as well as they are not directly related.
Quality:
1. The experimental results clearly validate the disentanglement and self-supervised learning quality as they improve the existing SSL baselines by large margins.
2. However, my main concern is about the justification part (Section 4) of the paper, where I am not fully convinced by the proofs of Lemma 1 (and therefore Theorem 1). In Line 155, the paper states there are two possible cases if Lemma 1 doesn't hold. But why are there only these two cases? It looks like the line 155-156 and the appendix line 61-62 try to describe the same thing: (1) main paper line 155-156: 'f is a collapsed representation that maps all inputs to a fixed vector, which is impossible in today's deep model training [88]'; (2) appendix line 61-62: 'The representation is not equivariant w.r.t. the action of \Pi G_i. As discussed in Lemma 1, this is impossible in today's deep model training [26].' However, nothing in the Lemma 1 discussion is related to the appendix statement about the equivariance with the product of subgroups, neither does the cited paper ([26] of the appendix). This makes me very confused about the correctness of the proof B.1 in the appendix.
In main paper lines 156-159, the paper cites [35] to say there is always a linear projection. But why is it related to the group G_{t+1}? And why is it related to the proof of Lemma 1?
If the SB-disentanglement is actually realized as claimed in section 4, can the authors show the specific learned groups (or subgroups) empirically? As long as I can see in the disentanglement experiments, what the paper tries to emphasize is the separation of latent subspaces, which corresponds to traditional disentangled representation learning without considering the groups that can actually act on them.
3. The unsupervised disentanglement experiment only reports the means of the metrics and no standard deviations. Typically models should be reported with runs of multiple random seeds to get an evaluation on disentanglement performance.
Clarity:
There could be some improvements.
1. There should be some preliminaries about IRM. At least a general introduction about what problem IRM is trying to solve, why it works, and why the SSL task falls into its setting.
2. The justification part (Section 4) should be improved on its proof of Lemma 1 and Theorem 1 (see my concerns in the Quality part).
Significance:
The direction of improving the disentanglement property of self-supervised learning representations is promising and important. The results are significant and I believe it will benefit the representation learning community.

**Time Spent Reviewing:**

8

---

> ### Author Response · Authors · 2021-08-10
> **Response to Reviewer 2DAu**
>
> Thanks for finding our work novel, promising, and important to the representation learning community. We will address all questions below.
>
> &nbsp;
>
> **Q1. Concerns on Lemma 1.**
>
> Sorry for the confusion. We will make it clear in the revision. Lemma 1 states that among entangled groups, there exists $\mathcal{G}\_{t+1}$ such that 1) $f$ is equivariant under the group action of $\mathcal{G}\_{t+1}$; 2) The group action of $\mathcal{G}\_{t+1}$ corresponds to linear maps in $\tilde{\mathcal{X}}\in \mathcal{X}$. Our justification is two-fold (hence the two cases in line 155):
>
> - *Equivariant*. When the semantic changes (i.e., different images), the feature changes.  It is reasonable because convolutional layers are equivariant [1]. This prevents the representation from mapping all inputs to a fixed vector (lines 155-156). We acknowledge that for a more theoretically complete solution, one needs to leverage specific models such as Normalizing Flow [2] or equivariant networks [3], which we will take as future works.
> - *Action of $\mathcal{G}\_{t+1}$ as linear map*. When $\mathcal{X}$ is a linear space, this clearly holds based on the group representation theory. When $\mathcal{X}$ is a non-linear manifold, [35] shows the existence of linear maps in a subspace of $\mathcal{X}$ that corresponds to at least one group, e.g., $\mathcal{G}\_{t+1}$ without loss of generality.
>
> Note that Appendix B.1 and B.2 are not proofs of the above two points. Instead, they detail the limitations of the standard SSL and supervised learning revealed by Lemma 1. Lemma 1 leads to Theorem 1: since the action of $\mathcal{G}\_{t+1}$ on $\tilde{\mathcal{X}} \in \mathcal{X}$ is preserved as linear maps, we can use the distance metrics in the linear space (e.g., cosine distance) to reflect the action of $\mathcal{G}\_{t+1}$. This enables us to use the maximization step in Theorem 1 to discover the partition of the dataset corresponding to $\mathcal{G}\_{t+1}$ (see lines 175-179), and we will elaborate this in Q2.
>
> &nbsp;
>
> **Q2. Show specific learned groups empirically.**
>
> The discovered partitions $\mathbf{P}^{*}$ after the maximization step for CMNIST dataset is shown in Figure 5 (a) of the main paper. We will use partition \#1 as an example, where the digit color is the same within each subset but different across subsets. This partition is related to the group $\mathcal{G}\_{t+1}$ acting on the digit color: the group action of $\mathcal{G}\_{t+1}$ is the identity mapping within each subset, but not identity mapping across subsets. Similarly, on real-world dataset STL10, the discovered partitions also correspond to specific semantics, e.g., grouping the images into ''has wing'' and ''no wing''. This discovered partition allows us to disentangle $\mathcal{G}\_{t+1}$ in the minimization step, i.e., decomposing a separate feature subspace $\mathcal{X}\_{t+1}$ affected only by $\mathcal{G}\_{t+1}$ to isolate the effect from $\mathcal{G}_{t+1}$ and achieve invariance across subsets (see lines 180-194).
>
> &nbsp;
>
> **Q3. Comparison with traditional disentangled representation learning.**
>
> Traditional disentanglement methods aim to separate the factors of variations by learning decomposed feature subspace. As far as the goal goes, this is indeed similar to our method (see lines 56-58). However, traditional methods adopt a statistical notion by assuming independence between factors, which is shown to be impossible to achieve disentanglement [4]. Furthermore, traditional methods rely on generative models such as VAE to learn the latent factors, which does not scale well to large real-world datasets such as ImageNet. In contrast, our IP-IRM is based on the group-theoretical definition proposed by Higgins [5] and does not make the independence assumption. It converges to a disentangled representation with theoretical guarantee without relying on VAE model, hence our method outperforms on small- to large-scale datasets.
>
> &nbsp;
>
> **Q4. Standard deviations of unsupervised disentanglement.**
>
> Thanks for your comments. We have added the standard deviations (4 trails) for the unsupervised disentanglement task on CMNIST dataset in the following table. The full results will be derived later and added in the final paper (due to the rebuttal time limitation).
>
> |      Methods      |         DCI         | IRS  |         MOD         |         EXP         |         LR          |         GBT         |       Average       |
> | :---------------: | :-----------------: | :--: | :-----------------: | :-----------------: | :-----------------: | :-----------------: | :-----------------: |
> |        VAE        | **0.948**$\pm$0.004 |  -   |   0.664$\pm$0.121   |   0.968$\pm$0.007   |   0.824$\pm$0.019   | **0.948**$\pm$0.004 |   0.849$\pm$0.057   |
> |    $\beta$-VAE    |   0.945$\pm$0.002   |  -   |   0.705$\pm$0.073   |   0.963$\pm$0.006   |   0.809$\pm$0.013   |   0.945$\pm$0.003   |   0.874$\pm$0.015   |
> | $\beta$-AnnealVAE |   0.911$\pm$0.002   |  -   |   0.790$\pm$0.075   |   0.965$\pm$0.007   |   0.821$\pm$0.022   |   0.911$\pm$0.002   |   0.880$\pm$0.015   |
> |   $\beta$-TCVAE   |   0.914$\pm$0.008   |  -   |   0.864$\pm$0.095   |   0.962$\pm$0.010   |   0.801$\pm$0.024   |   0.914$\pm$0.008   |   0.891$\pm$0.014   |
> |    Factor-VAE     |   0.916$\pm$0.004   |  -   | **0.893**$\pm$0.056 |   0.947$\pm$0.011   |   0.770$\pm$0.025   |   0.916$\pm$0.005   |   0.888$\pm$0.014   |
> |      SimCLR       |   0.882$\pm$0.019   |  -   |   0.767$\pm$0.025   |   0.976$\pm$0.011   |   0.863$\pm$0.036   |   0.876$\pm$0.015   |   0.873$\pm$0.016   |
> |   IP-IRM (Ours)   |   0.917$\pm$0.008   |  -   |   0.785$\pm$0.031   | **0.990**$\pm$0.002 | **0.921**$\pm$0.009 |   0.916$\pm$0.007   | **0.906**$\pm$0.011 |
>
> &nbsp;
>
> **Q5. Preliminaries about IRM.**
>
> Sorry for missing preliminaries about IRM. Here we give a brief introduction of IRM and will elaborate it in the final version.
>
> Invariant Risk Minimization (IRM) [6] is originally proposed to solve the Out-Of-Distribution (OOD) problem. It aims to find an invariant image representation through multiple training environments, thus robust towards unseen distributions. The core idea of IRM is to add an invariant risk regularization term to construct a predictor that performs equally well across different environments, i.e., regularizes model to be equivalently optimal in each environment.
>
> *Why we applied IRM term in SSL?* In [6], the author proved that the IRM term can be applied to any convex loss in a theoretical perspective. Besides, the cross entropy loss (in IRM) is essentially similar to the contrastive loss used for SSL. Please check Appendix B.1 and B.2 where we compare the cross entropy loss (supervised learning) and the contrastive loss (SSL) in detail. In our IP-IRM, we incorporate the IRM term into the SSL framework to disentangle semantics, by explicitly regularizing the model to be invariant across subsets. Please kindly refer to lines 175-196 in our main paper for more details.
>
> &nbsp;
>
> Reference:
>
> [1] Bronstein, Michael M., et al. "Geometric deep learning: Grids, groups, graphs, geodesics, and gauges." arXiv preprint arXiv:2104.13478 (2021).
>
> [2] Rezende, Danilo, and Shakir Mohamed. "Variational inference with normalizing flows." International conference on machine learning. PMLR, 2015.
>
> [3] Cohen, Taco, and Max Welling. "Group equivariant convolutional networks." International conference on machine learning. PMLR, 2016.
>
> [4] Locatello, Francesco, et al. "Challenging common assumptions in the unsupervised learning of disentangled representations." international conference on machine learning. PMLR, 2019.
>
> [5] Higgins, Irina, et al. "Towards a definition of disentangled representations." arXiv preprint arXiv:1812.02230 (2018).
>
> [6] Arjovsky, Martin, et al. "Invariant risk minimization." arXiv preprint arXiv:1907.02893 (2019).

---

> > ### Comment · Reviewer_2DAu · 2021-08-19
> > **After rebuttal comment**
> >
> > Thanks for the authors' response. I find the response informative, and I like this paper. I will raise my rating as accept.

---

### Official Review · Reviewer_Z5Cr · 2021-07-16

**Rating:** 7
**Confidence:** 3

**Summary:**

In this paper, the authors focus on the problem of learning unsupervised representations that are also disentangled. To this end, they build on top of the idea of invariant risk minimization and propose Iterative Partition-based Invariant Risk Minimization (IP-IRM) which iteratively assigns examples to partitions and constrains invariance in the loss among these partitions. They justify their approach mathematically according to a formal definition of disentanglement. The then perform multiple experiments qualitatively and quantitatively demonstrating the learned representations are disentangled. Furthermore, they show promising results on Out-of-distribution classification tasks.

**Ethical Concerns:**

I see no ethical concerns with this paper.

**Limitations And Societal Impact:**

Yes, they have addressed limitations and social impacts.

**Main Review:**

Strengths:

1. The paper is well written, flows well, and is easy to understand.

2. The problem is very relevant. There has been a great deal of work on learning self-supervised representations in computer vision. However, much work has focused on improving image classification numbers and less on how well the representation is disentangled. Disentanglement is an essential foundation for abstract reasoning.

3. The paper is very well motivated mathematically and conceptually. They show, using a formal definition of disentanglement, why IP-IRM is a sensible approach.

4. Quantitative and qualitative experimentation is very thorough and convincing.

Weaknesses:

1. For the OOD experiments on Imagenet, NICO, and related, the improvements in performance over the state of the art is smaller.

Overall:

The paper addresses an important but underexplored topic in self-supervised learning: disentangled representations. The authors firmly justify IP-IRM mathematically with Higgins's definition of disentanglement. Evaluation of the approach is thorough, and the quantitative and qualitative results convincing. I believe this would be a good contribution to the conference.

**Time Spent Reviewing:**

1 Hour

---

> ### Author Response · Authors · 2021-08-10
> **Response to Reviewer Z5Cr**
>
> Thanks for finding our paper well-motivated, well-written, and focused an important problem. We will address the weaknesses below.
>
> &nbsp;
>
> **Q1. The incremental improvements of OOD experiments.**
>
> Thanks for your comment. We gracefully disagree that our performance improvements are incremental. For example, we achieved about a margin of 3% on the NICO dataset. Moreover, as we mentioned in Appendix lines 477-491, if applying the existing deconfounding model [1] to the learned representation (using IP-IRM), we can achieve more improvements, e.g., about 8% in few-shot learning task as shown in Table E.4 of Appendix.
>
> &nbsp;
>
> Reference:
>
> [1] Yue, Zhongqi, et al. "Interventional few-shot learning." Advances in Neural Information Processing Systems, 2020.

---

> > ### Comment · Reviewer_Z5Cr · 2021-08-20
> > **Reponse to authors**
> >
> > Thank you for your informative response. I stand by my rating and still believe this paper would be a good contribution to the conference.

---

### Official Review · Reviewer_4t1m · 2021-07-16

**Rating:** 9
**Confidence:** 4

**Summary:**

This paper proposes a scalable approach to disentangled representation learning using the self supervised learning framework, rather than the typically used generative model framework. Their approach is principled, being grounded in the Higgins et al definition of disentangling from the symmetry perspective. It also appears to be scalable, which is a very exciting step in the subfield of unsupervised disentangled representation learning.

**Main Review:**

The paper is very well written and makes a very exciting step in unsupervised disentangled representation learning. I am happy to argue for its acceptance. Saying this, I do have a few concerns.

1) Is K a hyperparameter? How should one choose it? What were the choices for CMNIST and 3D shapes? Why does increasing it result in reduced performance? How do the authors propose to scale their algorithm to disentangling many different subspaces without the ability to increase K?

2) What are the augmentations used, and how are the negative examples chosen? If the authors just augment the existing SSL algorithms with their native augmentations/negative examples with the proposed disentangling framework, then it should be stated in text, as otherwise the reader is left wondering. Is there a way to choose augmentations that would not interfere with disentangling, as the authors suggested e.g. that colour information was not disentangled because it was subsumed by the augmentations.

Minor: Line 147 "In particular, SSL and fully-supervised learning are two special cases of Gi-disentanglement" - I think it would be better to move it out of the lemma, as this is not really a part of it, and instead add this to the following more in depth discussion of this point later in the section.





**Time Spent Reviewing:**

2

---

> ### Author Response · Authors · 2021-08-10
> **Response to Reviewer 4t1m**
>
> Thanks for finding our work very exciting and interesting. We will revise Lemma 1 as suggested and address other questions in the revision.
>
> &nbsp;
>
> **Q1. Hyperparameter $K$ and its impact on performance.**
>
> As you pointed out, $K$ is a hyperparameter. We fixed it as $2$ in all of our main experiments, except the ablation study that we did in Figure 6 (a) to evaluate the SSL performance with varying $K$. We presented the detailed results in Appendix lines 258-269. Briefly, increasing $K$ reduces \# negative samples in each subset. SSL performance is sensitive to \# negative samples, so it decreases (when increasing $K$). Note that the value of $K$ does not affect the disentanglement capability of IP-IRM and $K=2$ suffices to achieve full disentanglement (please check the proof in Appendix B.3) by iterating Step 1-2 in Section 3.
>
> &nbsp;
>
> **Q2. Augmentations and negative examples in IP-IRM.**
>
> Sorry for the confusion. As you pointed out, we followed the existing SSL frameworks (e.g., SimCLR) regarding the augmentations/negative examples. We will highlight this in the revision. Regarding the augmentations, one can manually select augmentations that do not alter the data-generating semantics on disentanglement datasets generated from a few semantics (e.g., 3dshapes dataset), as in recent work [1]. On real-world datasets such as ImageNet, one can just follow the standard augmentations used in SSL methods. Due to the high complexity and quantity of the semantics in real datasets, it is very rare that a few of standard augmentations can alter them.
>
> &nbsp;
>
> Reference:
>
> [1] von Kügelgen, Julius, et al. "Self-Supervised Learning with Data Augmentations Provably Isolates Content from Style." arXiv preprint arXiv:2106.04619 (2021).

---

> > ### Comment · Reviewer_4t1m · 2021-08-19
> > **Thank you**
> >
> > Thank you for your response. My score was high to begin with, so I will leave it unchanged. Good work!

---

### Official Review · Reviewer_nijM · 2021-07-19

**Rating:** 7
**Confidence:** 3

**Summary:**

This paper presents a new self-supervised learning technique that learns to map image inputs to "disentangled" vectors iteratively by alternating two steps. First, all of the data is assumed to be in the same subset and they learn a representation by minimizing a SimCLR like contrastive loss (here the positive pairs are augmented versions of the same image, while negative samples are other images in the batch). In the second step, they find the partition of the dataset into 2 subsets that maximize the contrastive loss. This new partition is added to the set of all partitions considered, and we repeat the two steps (again learning the representation that minimizes contrastive loss over all partitions and then finding the next partition and so on). They use a regularized form of the contrastive loss, which adds a term taken from earlier work ("Invariation Risk Minimization"). This terms measures the magnitude of the gradient of the contrastive loss wrt to the temperature parameter (at temperature=1). They evaluate their technique against alternatives on a wide range of disentanglement metrics and downstream classification accuracy, and show their technique performs well.

**Ethical Concerns:**

No ethical concerns.

**Limitations And Societal Impact:**

Yes.

**Main Review:**

Overall, I think the paper is quite interesting and should be of interest to the community. I especially like the idea of finding the worst partition in terms of the representation learned so far and re-learning the representation based on this partition. I also found the evaluations in the paper quite comprehensive.

I think the main issue with the paper is that it makes rather strong claims but fails to support some of these. For example, the authors argue that their technique can disentangle the full semantic space (i.e., learn a fully disentangled representation) in a fully unsupervised manner. (in contrast to other SSL techniques, which can only learn to disentangle the semantic information wrt hand-crafted augmentations.) However, to me it is not clear how this squares with the Locatello et al. result that shows that fully unsupervised learning of disentangled representations is not possible. They mention in passing that Higgins et al.'s group theoretic definition of disentanglement solves the problem with Locatello et al.'s result but this is not clear to me. Locatello et al. in essence shows that there are multiple equivalent representations and it is impossible to pick one over another as more disentangled without some supervision. And Higgins et al. also accept this fact as they point out in their paper that their disentanglement definition assumes a "natural" decomposition is already given (i.e., what it means to be disentangled). I should say that I found it difficult to fully grasp what the authors' claim is, so please let me know if I misunderstood anything.

This brings me to my other point. I think the paper right now is unfortunately hard to read. The paper gets rather technical at times, and it is hard to follow the argument (for example, discussion after Lemma 1 in Section 4 is hard to follow). While I appreciate the authors' presenting theoretical results, I think the paper would benefit greatly if the technique could be better motivated on an intuitive level, and the technical arguments similarly could be presented on an intuitive level.

Please also see my further comments below:
- Figure 2 is hard to understand. What is being classified on the left panel? Also, it is not clear what I should see on the heatmaps on the right.
- line 105-106, the authors say "Higgins et al., ..., resolves the previous controversial points" wrt to Locatello et al. for example. As I mentioned above, this is not clear to me. It'd be worth expanding this a little bit further.
- What is the effect of regularizer term from "Invariant Risk Maximization"? The authors say it "regularizes phi to be invariant across subset in a partition". What does this mean exactly? Can you give an intuitive example of what this term does? What would happen if you didn't have this term?
- line 155, the authors say "f is a collapsed representation that maps all inputs to a fixed vector, which is impossible in today’s deep model training". What does this mean? It is impossible because of the regularization or tricks like EMA that people use. I don't see how the referenced work [88] is relevant to this point either.
- To me it seemed like finding the partition that maximizes loss is akin to finding hard negatives. Is this intuition correct? If so, would it be useful to mention sth along these lines in the paper?
- For the results in Table 2, how do you combine IP-IRM with SimCLR etc.?
- In the checklist, the authors mention limitations are discussed in section 6, but I couldn't see this.
- I understand that the authors may not have had enough time to get standard deviations on all of the results, but it'd be nice to add these. (I have seen the results in appendix).
- While reading this paper, I was reminded of the paper "What Should Not Be Contrastive in Contrastive Learning" ()https://arxiv.org/abs/2008.05659). This paper learns essentially a separate subspace for each augmentation (which encourages the representations not to throw away information like SimCLR etc. might do). It'd be interesting to see how well their method compares to this. (To be clear, this is totally optional. I understand the authors might not have time to work on this.)
- typos:
  - line 74, "attributed first part" -> "attributed to first part"
  - line 218, "regrading"
  - line 228, "What does IP-IRM feature look like?" -> "What do IP-IRM features look like?"
  - line 239, "regardless the other semantic" -> "regardless of the other semantic"



**Time Spent Reviewing:**

3

---

> ### Author Response · Authors · 2021-08-10
> **Response to Reviewer nijM [2/2]**
>
> **Q8. Comparison with ``What Should Not Be Contrastive in Contrastive Learning'' [5].**
>
> Thanks for your constructive advice. We have carefully read the paper and reproduced the proposed LooC [5] based on SimCLR for fair comparison.
>
> As you pointed out, LooC can disentangle each augmentation by constructing separate embedding spaces. This is a kind of manual intervention as a special case of our IP-IRM (automatically discovery the partition). From table below, we can find that LooC improves SimCLR which only guarantee to disentangle a single feature subspace affected by all augmentations, which validates the correctness of our theory. However, LooC still has no guarantee to disentangle beyond augmentations (e.g., other semantics). Table shows that all the IP-IRM combination methods outperform the LooC. Specifically, HCL+IP-IRM can outperform it by 3.02% for $k$-NN accuracy on CIFAR100 dataset.
>
> |    Methods    |   k-NN    |  Linear   |
> | :-----------: | :-------: | :-------: |
> |    SimCLR     |   54.94   |   66.63   |
> |      DCL      |   57.29   |   68.59   |
> |      HCL      |   59.61   |   69.22   |
> |     LooC      |   57.03   |   68.61   |
> | SimCLR+IP-IRM |   59.10   |   69.55   |
> |  DCL+IP-IRM   |   58.37   |   68.76   |
> |  HCL+IP-IRM   | **60.05** | **69.95** |
>
>
> &nbsp;
> &nbsp;
>
> Reference:
>
> [1] Arjovsky, Martin, et al. "Invariant risk minimization." arXiv preprint arXiv:1907.02893 (2019).
>
> [2] Grill, Jean-Bastien, et al. "Bootstrap your own latent: A new approach to self-supervised learning." arXiv preprint arXiv:2006.07733 (2020).
>
> [3] Bronstein, Michael M., et al. "Geometric deep learning: Grids, groups, graphs, geodesics, and gauges." arXiv preprint arXiv:2104.13478 (2021).
>
> [4] Chen, Ting, et al. "A simple framework for contrastive learning of visual representations." International conference on machine learning. PMLR, 2020.
>
> [5] Xiao, Tete, et al. "What should not be contrastive in contrastive learning." arXiv preprint arXiv:2008.05659 (2020).

---

> > ### Comment · Reviewer_nijM · 2021-08-23
> > **Thanks for the response**
> >
> > I'd like to thank the authors for their detailed response, which addresses all of my concerns. I think this is a very interesting paper and should be of interest to the whole community.

---

> ### Author Response · Authors · 2021-08-10
> **Response to Reviewer nijM [1/2]**
>
> Thank you for finding our work interesting and valuable to the community. We apologize for the typos and will fix them in revision. We will address all your concerns/comments below:
>
> &nbsp;
>
> **Q1. Learning fully disentangled representation.**
>
> Sorry for the clarity issue on learning fully disentangled representation.
>
> Thanks for pointing this out. We agree that both Locatello et al. and Higgins et al. *indeed* point out the impossibility of unsupervised learning of disentangled representation using a statistical approach, i.e., discovering independent latent factors (e.g., VAE). To address the limitation of the statistical approach towards disentanglement, Higgins et al. propose a group theoretical definition by assuming a natural decomposition that reflects the structure of the world. They point out (in Section 3, page 8 of their paper) that it is *possible* to achieve disentanglement w.r.t. the natural decomposition through active intervention. Yet, the approach of such intervention is not discussed in Higgins' work. Our IP-IRM actually offers a practical solution under Higgins' definition. IP-IRM learns disentangled representation without supervision by discovering semantic partitions through *active intervention*. Specifically, it achieves the partition in an optimizable way by continuously exchanging samples between subsets (see Eq.(3)). The resulting partition corresponds to an intervention through $\mathcal{G}\_{t+1}$: the action of $\mathcal{G}\_{t+1}$ is the identity mapping within each subset, while not identity mapping across subsets. Then minimizing Eq.(2) with the obtained partition leads to a $\mathcal{G}_{t+1}$-disentangled representation. Please refer to Q2 for more intuitive explanations. We will definitely improve the clarity of our idea in the revision.
>
> &nbsp;
>
> **Q2. Intuition of the IRM regularization term and our IP-IRM.**
>
> The IRM regularization term [1] was originally proposed to solve the Out-Of-Distribution (OOD) problem in classification tasks based on the cross entropy loss. The core idea is to regularize the classification model to be equivalently optimal across all given data distributions. For example, given a task of recognizing digit ''0'' and ''1'', let the training data contain two distributions of digit images: one with most ''0'' in red, most ''1'' in green; and the other one with balanced color for ''0'' and ''1''. Using IRM regularization term makes the model be invariant to the spurious color-digit correlation and focus on only the features that distinguish digits such as shape. Please check [1] for more details.
>
> Similarly, in our IP-IRM, the IRM regularization term regularizes the model $\phi$ to be invariant across subsets (see lines 126-128). Overall, our IP-IRM has 2 steps: maximization step aims to discover a group $\mathcal{G}\_{t+1}$ which is still entangled; while minimization step implements a partition-based SSL with the IRM regularization term to disentangle the representation w.r.t $\mathcal{G}\_{t+1}$.
>
> - *Maximization Step*. As you pointed out, the maximization step in IP-IRM can be intuitively understood as finding hard negatives. This is achieved by discovering a partition based on group $\mathcal{G}\_{t+1}$ which is still entangled. In fact, Lemma 1 explains why the partition corresponds to hard negatives: the partition maximizes the sample similarity in each subset, making SSL more difficult within each subset (see lines 175-179 and Figure 3). In this way, the action of $\mathcal{G}\_{t+1}$ is the identity mapping within each subset, but not identity mapping across different subsets. For example, as shown in Figure 5 (a) Partition \#1, the discovered partition is based on the group $\mathcal{G}\_{t+1}$ acting on the *color* semantic: subset \#1 is almost red while \#2 is green.
> - *Minimization Step*. Maximization step and minimization steps work alternatively. After the above maximization step, we have the minimization step to apply the IRM regularization term with the obtained partition. To achieve invariance between subset \#1 (red digits) and \#2 (green digits), the model learns to decompose a separate feature subspace $\mathcal{X}\_{t+1}$ to encode the *color* semantic, thus isolate the effect of color across different subsets. In this way, color semantic is disentangled (see lines 180-194 for details).
>
> We will include more intuitive explanations about IRM and our IP-IRM in revision.
>
> &nbsp;
>
> **Q3. Clarity of Figure 2.**
>
> We will address the clarity issue of Figure 2 in the revision.
>
> On the left, we compare the representation learned by SimCLR and our IP-IRM on STL10 dataset. We report the $k$-NN classification accuracy using the aug. equivariant feature and aug. unrelated feature, respectively. Please check more elaboration on lines 71-75.
>
> On the right, we show the image semantics captured by the filters of the CNN backbone. We aim to show that using our IP-IRM can disentangle more semantics than using standard SSL methods (e.g., SimCLR). For example for the rightmost example (a building image), IP-IRM representation at 29th layer roughly disentangles the rectangular building, the stairs, the cylindrical monument and the side buildings into different feature maps, while SimCLR entangles them.
>
> &nbsp;
>
> **Q4. Impossibility of collapsed representation.**
>
> Sorry for the confusion. Here we give a more detailed explanation and will make it clearer in revision.
>
> In our work, the collapsed representation means two-fold: 1) *Model $\phi$ maps all inputs to a fixed representation*. As you pointed out, this is impossible because the contrasting negative pairs of the contrastive loss (used in SimCLR) as well as the tricks like EMA (used in BYOL [2]) prevent the representation collapse.  2) *Model $\phi$ maps each input to an isolated vector.* This is impossible due to the equivariant deep neural network architecture (e.g., the CNN convolutional layers of the backbone [3]). The equivariance preserves the group structure, thus prevents from mapping each input to an isolated vector (e.g., random mapping).
>
> &nbsp;
>
> **Q5. How to combine IP-IRM with SimCLR in Table 2.**
>
> For implementation, we just replace the loss function $\mathcal{L}$ in Eq.(2) and (3) of the main paper with the SimCLR loss function (Eq.(1) in [4]).
>
> &nbsp;
>
> **Q6. Discussion on limitations.**
>
> Sorry for the confusion, we mention how IP-IRM can be improved in Section 6, and we will elaborate its limitation in revision. The main limitation of IP-IRM is the convergence speed. We can see this from Eq.(2), where the representation is trained w.r.t. all the discovered partitions. However, this means that the IP-IRM model receives less training w.r.t. a partition discovered at a later timestamp compared to ones at an earlier timestamp. This can limit the performance of the proposed approach, as our IP-IRM is based on SSL, whose performance is sensitive to the \#training epochs. Hence future works can focus on improving the convergence speed to alleviate this issue.
>
> &nbsp;
>
> **Q7. Standard deviations.**
>
> Thanks for your comments. We have added the standard deviations (4 trails) for the unsupervised disentanglement task on CMNIST dataset in the following table. The full results will be derived later and added in the final paper (due to the rebuttal time limitation).
>
> |      Methods      |         DCI         | IRS  |         MOD         |         EXP         |         LR          |         GBT         |       Average       |
> | :---------------: | :-----------------: | :--: | :-----------------: | :-----------------: | :-----------------: | :-----------------: | :-----------------: |
> |        VAE        | **0.948**$\pm$0.004 |  -   |   0.664$\pm$0.121   |   0.968$\pm$0.007   |   0.824$\pm$0.019   | **0.948**$\pm$0.004 |   0.849$\pm$0.057   |
> |    $\beta$-VAE    |   0.945$\pm$0.002   |  -   |   0.705$\pm$0.073   |   0.963$\pm$0.006   |   0.809$\pm$0.013   |   0.945$\pm$0.003   |   0.874$\pm$0.015   |
> | $\beta$-AnnealVAE |   0.911$\pm$0.002   |  -   |   0.790$\pm$0.075   |   0.965$\pm$0.007   |   0.821$\pm$0.022   |   0.911$\pm$0.002   |   0.880$\pm$0.016   |
> |   $\beta$-TCVAE   |   0.914$\pm$0.008   |  -   |   0.864$\pm$0.095   |   0.962$\pm$0.010   |   0.801$\pm$0.024   |   0.914$\pm$0.008   |   0.891$\pm$0.014   |
> |    Factor-VAE     |   0.916$\pm$0.004   |  -   | **0.893**$\pm$0.056 |   0.947$\pm$0.011   |   0.770$\pm$0.025   |   0.916$\pm$0.005   |   0.888$\pm$0.014   |
> |      SimCLR       |   0.882$\pm$0.019   |  -   |   0.767$\pm$0.025   |   0.976$\pm$0.011   |   0.863$\pm$0.036   |   0.876$\pm$0.015   |   0.873$\pm$0.016   |
> |   IP-IRM (Ours)   |   0.917$\pm$0.008   |  -   |   0.785$\pm$0.031   | **0.990**$\pm$0.002 | **0.921**$\pm$0.009 |   0.916$\pm$0.007   | **0.906**$\pm$0.011 |

---

### Author Response · Authors · 2021-08-24
**To all reviewers**

We thank all the reviewers for their positive feedback and constructive suggestions. We are very happy that all the reviewers find our response detailed and informative, as well as agree that our work should be of interest and a good contribution to the whole community. We will carefully revise our paper by taking into account all the suggestions and concerns.

---

### Public Comment · ~Yivan_Zhang1 · 2022-02-18
**Minor definition issues**

Thanks for this nice paper! I like this formulation.

Although, I found some minor issues about the definition:

- A **group action**'s domain and codomain are $\cdot: \mathcal{G} \times \mathcal{U} \to \mathcal{U}$, so it should be $g \cdot u \in \mathcal{U}$, not $\mathcal{U} \times \mathcal{U}$ (or we can say $g \cdot (-): \mathcal{U} \to \mathcal{U}$ is an automorphism that transforms $u$);
- A **group representation** of a group $\mathcal{G}$ on a set $\mathcal{X}$ is a mapping from the set of group elements to the set of automorphisms of $\mathcal{X}$ (which is a subset of all endomorphisms $\mathcal{X}^\mathcal{X}$) that preserves the identity, composition, and inverse. In other words, it is a group homomorphism $\mathcal{G} \to Aut(\mathcal{X})$ from the group $\mathcal{G}$ to the automorphism group $Aut(\mathcal{X})$ (when $\mathcal{X}$ is a vector space, the general linear group $GL(\mathcal{X})$), not $\mathcal{G} \to \mathcal{X} \times \mathcal{X}$.

The current writing may be a bit misleading. I hope this information is also helpful for other readers.

Best regards,
Yivan

---

> ### Public Comment · ~Zhongqi_Yue1 · 2022-02-25
> **Reply to Yivan**
>
> Thanks for your interest and suggestion! We will look into this in the next revision. In the current version, we have included a detailed preliminaries section in appendix on group action and representation.

---

> > ### Public Comment · ~Yivan_Zhang1 · 2022-03-01
> > **Thanks**
> >
> > Thanks for your reply.
> >
> > I checked the appendix. The definitions there are correct. Then it's just a minor notation issue in Definition 1 and the paragraph below it.
> >
> > Best,
> > Yivan

---

### Decision · Program_Chairs · 2021-09-27

**Decision:**

Accept (Spotlight)

**Comment:**

This paper proposes a novel approach to an important and popular problem, and in doing so provides a natural but novel application of *another* important and popular line of work (IRM). The work is more theoretically justified than most in the area, and has reasonably satisfying experimental results. Overall, it will make a fine contribution to the conference. To maximize that, please make sure to incorporate the outcome of the reviewer discussions – some of which seemed fairly illuminating – into the final version of the paper.